# Mind Control through Causal Inference: Predicting Clean Images from Poisoned Data

**Mengxuan Hu**[1*], **Zihan Guan**[1*], **Yi Zeng**[2] , **Junfeng Guo**[3] , **Zhongliang Zhou**[4], **Jielu Zhang**[5],
**Ruoxi Jia**[2], **Anil Vullikanti**[1†], **Sheng Li**[1†]
[1]University of Virginia
[2]Virginia Tech
[3]University of Maryland, College Park
[4]Merck & Co., Inc.
[5]University of Georgia
[*]These authors contributed equally
[†]Co-corresponding Author

## Abstract

*Anti-backdoor learning*, aiming to train clean models directly from poisoned datasets, serves as an important defense method for backdoor attack. While existing methods can prevent models from predicting the target label on backdoored samples, they face significant challenges in **recovering** backdoored samples to their original, correct labels. Additionally, their non-end-to-end training frameworks make them unsuitable for safeguarding the increasingly prevalent large pre-trained models. To bridge the gap, we first revisit the anti-backdoor learning problem from a causal perspective. Our theoretical causal analysis reveals that incorporating ***both*** images and the associated attack indicators preserves the model's integrity. Building on the theoretical analysis, we introduce an end-to-end method, Mind Control through Causal Inference (MCCI), to train clean models directly from poisoned datasets. This approach leverages both the image and the attack indicator to train the model. Based on this training paradigm, the model's perception of whether an input is clean or backdoored can be controlled. Typically, by introducing fake non-attack indicators, the model perceives all inputs as clean and makes correct predictions, even for poisoned samples. Extensive experiments demonstrate that our model can effectively and robustly recover the **original true labels** of backdoored images, without compromising clean accuracy. Our code can be found at https://github.com/xuanxuan03021/BKD_BKD_ICLR.

## 1 Introduction

While deep neural networks (DNNs) have achieved tremendous success in various fields (Kortli et al., 2020; Zou et al., 2023; Vaswani et al., 2017; Sun et al., 2022), training these models requires a massive amount of data. Consequently, it has become common practice to outsource the collection of training data to third-party providers, especially for large pre-trained models. For example, the large foundation model CLIP is trained over millions of image-text pairs, which are directly collected from the internet. However, this practice opens the door for malicious attackers to manipulate model behavior by poisoning the training dataset. This type of manipulation is known as *backdoor attacks*, where an attacker poisons the training dataset so that DNNs trained on this dataset will classify images to a target label when a specific trigger is present in the input, but behave normally with benign inputs. Backdoor attacks pose a serious security vulnerability to DNNs, especially in safety-critical scenarios such as autonomous driving (Han et al., 2022; Chan et al., 2022), medical diagnosis (Feng et al., 2022), and financial fraud detection (Lunghi et al., 2023).

Hence, this raises an inspiring yet challenging research question: *How can we directly train a backdoor-free model even when backdoor samples are hidden in the training dataset*? Despite a few prior endeavors (Li et al., 2021a; Zhang et al., 2023; Huang et al., 2022) on this problem, several crucial challenges remain unresolved for this scenario. ❶ **Ignorant Label-Recovering Ability**:

Most existing methods primarily focus on preventing a backdoor-free model from predicting the target label for backdoored samples, while neglecting the practical need to recover the original, correct predictions for such samples. As evidenced in (Wu et al., 2022), all the existing anti-backdoor learning methods fail to consistently achieve strong recovery performance across various backdoor attacks. ❷ **Low Generalization To Large Pre-trained Models**: Existing approaches (Li et al., 2021a; Zhang et al., 2023; Huang et al., 2022; Gao et al., 2023; Zhu et al., 2023) involve complex multi-stage training processes that significantly increase computational costs. Particularly during the initial stage, these methods often isolate potential backdoored and clean samples by training the victim model on the entire training set, which can be computationally expensive, especially for large pre-trained models. For instance, vision foundation models like CLIP and BLIP, which contain approximately 80 million parameters and use training sets of around 400 million image-text pairs, incur substantial isolation costs. ❸ **High Dependence on Backdoor Isolation Performance**: From a technical point of view, the existing methods rely heavily on the efficacy of backdoor isolation. For instance, (Li et al., 2021a) unlearns embedded backdoors using isolated backdoor samples in the initial stage, (Zhang et al., 2021) learns a disentangled representation by isolating the backbone model from an intentionally-trained backdoored model, and (Huang et al., 2022; Gao et al., 2023) attempt to prevent backdoors by isolating low-credibility samples using the model itself. This reliance on backdoor isolation limits the models' ability to generalize effectively in scenarios where backdoors are difficult to detect and isolate. Therefore, an important question arises: *How can we efficiently train an end-to-end clean model on a poisoned dataset that not only maintains high accuracy on clean samples but also **recovers** the correct labels of backdoor samples?*

To address this question, we first compare the intrinsic differences between training a poisoned model and a clean model on a poisoned dataset from a causal perspective. Specifically, through rigorous theoretical derivation based on the **backdoor adjustment theory** in causal inference, we find that training a model solely with images from a poisoned dataset may compromise its integrity. Conversely, incorporating **both** images and associated attack indicators can safeguard the model's integrity. Typically, attack indicators represent the probability of the presence of an attack on the corresponding image. Inspired by (Zeng et al., 2021), we use the frequency spectrum of each image as the attack indicator. Based on this insight, we propose our end-to-end Mind Control through Causal Inference (MCCI) model, as illustrated in Fig. 1. MCCI utilizes the given victim model (e.g. ViT) as a Semantic Feature Recognition Network (SFRN) to learn semantic features directly from images. It also introduces a smaller Attack Indication Network (AIN) to augment the victim model. This AIN determines whether an input is backdoored given the attack indicator. Hence, it acts as a "magic wand", **controlling the model's perception** of whether an input is clean or backdoored. In the inference stage, we leverage counterfactual outcome estimation to manipulate the model's perception, making it treat all inputs as clean by introducing fake non-attack indicators to the AIN. Extensive experiments demonstrate that backdoored samples can be effectively and robustly recovered to their original labels without compromising clean accuracy. Furthermore, additional experiments uncover the mechanism behind the mind control secret of AIN through latent space visualization.

Our contributions are summarized as follows: **(1) Advanced Recovery Ability**. We addressed an overlooked aspect of evaluating the recovery ability of backdoor-free models on poisoned datasets. Compared to the existing baselines, our MCCI model has shown outstanding label recovery performance. **(2) Causality-Inspired Training and Inference**. We leverage theories in causal inference to analyze the differences between training a poisoned model and a clean model on a poisoned dataset and develop a counterfactual reasoning approach for inference-stage backdoor sample recovery. **(3) Generalization to Large Pre-trained Models**. We empirically demonstrate that MCCI is highly adaptable for backdoor defense in large pre-trained models with limited training overhead. **(4) SOTA Performance in Effectiveness and Efficiency**. Extensive experiments demonstrate that our method effectively and efficiently trains clean models on poisoned datasets.

## 2 RELATED WORK

**Backdoor Attack.** Backdoor attacks are usually launched through data poisoning (Gu et al., 2017; Chen et al., 2017; Nguyen & Tran, 2021; Li et al., 2021c; Liu et al., 2020; Shafahi et al., 2018a; Schneider et al., 2024; Li et al., 2024b; Lan et al., 2024; Cheng et al., 2024; Yin et al., 2024; Guan et al., 2023b; Guo et al., 2024), where malicious attackers inject backdoor samples into the training dataset. As a result, a model trained on this dataset will classify images to a target label when a

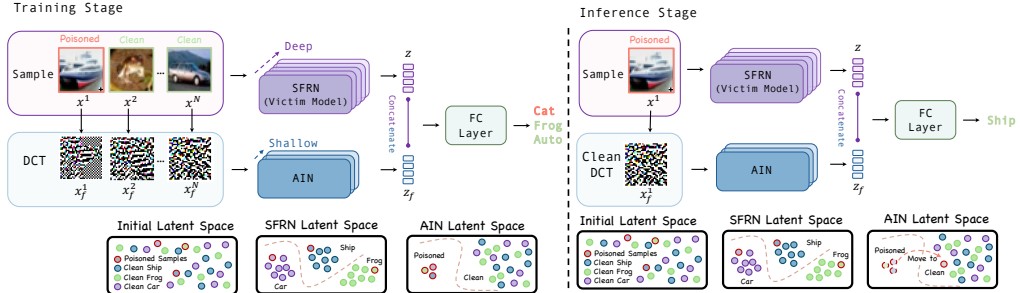

Figure 1: Pipeline of the MCCI. In the training stage, SFRN processes the original image $x$ to recognize complex semantic features (shown in the SFRN latent space) and encodes $x$ into $z$. Simultaneously, AIN analyzes the frequency spectrum $x_f$ of $x$ to assesses whether the input is backdoored (illustrated in the AIN latent space) and encodes $x_f$ into $z_f$. The two embeddings, $z$ and $z_f$, are then concatenated for the final image prediction. In the inference stage, a clean frequency spectrum is inputted along with the image to ensure clean predictions.

specific trigger is present in the input while maintaining normal behavior on benign inputs. Although there are other categories of backdoor attacks (Appendix I), this paper primarily focuses on data poisoning-based backdoor attacks.

**Backdoor Defense.** Following (Li et al., 2022b), the existing backdoor defense methods can be roughly split into five main categories, including (1) detection-based defenses (Huang et al., 2019; Xiang et al., 2022; Guo et al., 2021), (2) preprocessing-based defenses (Doan et al., 2020; Li et al., 2021b; Huang et al., 2023; Shi et al., 2024; Borgnia et al., 2021), (3) poison-suppression-based defense (Li et al., 2021a; Huang et al., 2022; Zhang et al., 2023; Gao et al., 2023), (4) trigger-synthesis-based defense (Wang et al., 2019; Chen et al., 2022; Xu et al., 2023), and (5) input-filtering-based defenses (Gao et al., 2019; Zeng et al., 2021; Guan et al., 2023a; Guo et al., 2023a;b; Wang et al., 2024; Hu et al., 2024; Guan et al., 2024). In particular, in this paper, we adopt a poison-suppression-based threat model similar to well-established methods (Li et al., 2021a). Few methods have been proposed in this research direction. Notable approaches include DBD and its variations (Huang et al., 2022; Chen et al., 2022), which decouples training between DNN backbone and fully connected layers to minimize trigger-label correlation; ABL (Li et al., 2021a) isolates and then unlearns backdoors through gradient ascent; V&B (Zhu et al., 2023) exploits backdoored model to erase potential backdoors, and CBD (Zhang et al., 2023), employing a causality-inspired structure to learn de-confounded representations for accurate classification.

# 3 PROBLEM SETUP

## 3.1 THREAT MODEL

In this paper, the defender aims to train a clean model end-to-end from a poisoned dataset with the following two objectives: ❶ the clean model is expected to produce the **original, correct predictions** for backdoored images even if the trigger is present, and ❷ the clean model is expected to produce high-accuracy predictions for the clean samples, achieving performance as good as the model trained on a purely clean dataset. ❸ The training process should be in an end-to-end fashion, which can easily generalize to large pretrained models with reasonable computational cost. We assume that the defender has full control of the training process, but lacks prior knowledge about the proportion and the distribution of backdoor samples in the poisoned dataset. Fig. 8 in the Appendix offers a visualization of the considered threat model.

**Comparison with the Past Threat Models.** Our defense goal is more challenging and practical compared to the threat models in (Li et al., 2021a; Zhang et al., 2023; Huang et al., 2022). While the defender in those threat models aims to train a clean model that *only* avoids misclassifying backdoored samples as the target label, our defense objective additionally expects the model to *predict the original, correct labels for backdoored samples in an end-to-end manner with high generalization to large pretrained models*. Therefore, this objective is more practical (TRC'22, 2022) in the

real world. Otherwise, the model may make unreasonable predictions (e.g., assigning random labels other than the target label) when exposed to backdoor trigger patterns, posing a serious threat to model security.

## 3.2 PRELIMINARIES

To uncover the intrinsic differences between training a poisoned model and a clean model on a poisoned dataset, we leverage causal inference for analysis. In particular, causal inference aims to estimate the causal effect of the treatment $T$ on the outcome $Y$. Analogous to the image classification task, our objective is to estimate the effect of an image on the prediction. Therefore, we treat the image as the treatment and its corresponding prediction as the outcome. We first introduce some basic definitions in causal inference, along with other assumptions (Yao et al., 2021; Guo et al., 2020; Pearl, 2009), causal effect can be identified from the observational data. More details are provided in the Appendix B.

**Definition 3.1** (Confounders). Confounders are variables that serve as a common cause of the treatment $T$ and the outcome $Y$.

**Assumption 3.2** (Positivity). Every unit should have non-zero probabilities to be assigned in each treatment group. Formally, $\mathbb{P}(T = t | X = x) \neq 0, \forall t \in \mathcal{T}, \forall x \in X$.

**Assumption 3.3** (Modularity). Given a node $x_i$ and its parent nodes $pa_i$ in the causal graph, if we intervene on a set of nodes $S$ by assigning them constants, then for all $i$: 1. If $i \notin S$, then $\mathbb{P}(x_i \mid pa_i)$ remains unchanged. 2. If $i \in S$ and $x_i$ is the value that $X_i$ was set to by the intervention, then $\mathbb{P}(x_i \mid pa_i) = 1$; otherwise, if $x_i$ is not the intervention value, $\mathbb{P}(x_i \mid pa_i) = 0$.

## 3.3 BACKDOOR ATTACK FORMATION FROM A CAUSAL VIEW

As preliminarily explored in (Zhang et al., 2023), the underlying mechanism of backdoor attack formation can be demonstrated using a causal graph, as shown in Fig. 2(a). Specifically, backdoor attacks $A$ are initiated by attaching a trigger to the victim image $A \rightarrow I$ and altering the image label $Y$ to the target label $A \rightarrow Y$. Hence, the confounder $A$ introduces a spurious backdoor path $I \leftarrow A \rightarrow Y$ that builds and strengthens erroneous correlations between the modified images and the tar-

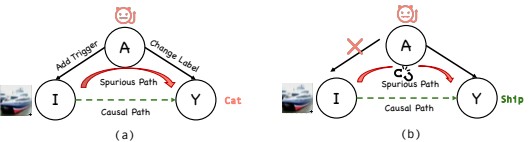

Figure 2: (a) Causal graph of traditional backdoor attack. (b) Causal graph of backdoor defense by cutting off the edge between $A \rightarrow I$ (backdoor adjustment).

get label, misleading the predictions of backdoor samples instead of guiding them to follow the real causal path $I \rightarrow Y$. Formally, in causal inference, this misled backdoor prediction is attributed to the **confounding bias** brought about by the confounder variable $A$. Therefore, the prediction solely based on the image $\mathbb{E}(Y|I = i)$ is inevitably influenced by this spurious correlation. Formally,

**Proposition 3.4.** *Let $i$ be an image from a poisoned dataset and $f_\theta$ a model trained on this dataset. Then, the expected prediction $Y$ of image $I = i$ can be derived according to Fig. 2:*

$$\mathbb{E}(Y|I = i) = \mathbb{E}_A[\mathbb{E}[Y|i, A] \underbrace{\frac{\mathbb{P}(i|A)}{\mathbb{P}(i)}}_{\text{Source of confounding bias}}] \tag{1}$$

*Proof.* We provide a detailed derivation in Appendix C.1.

## 4 MIND CONTROL THROUGH CAUSAL INFERENCE (MCCI)

In this section, we provide a detailed introduction to the training and inference stages of the MCCI model. We begin by discussing the causal theory used to eliminate confounding bias, then adapt this theory to the backdoor setting through an analogous theorem. Following this, we describe the MCCI training algorithm based on the theorem. Finally, we introduce counterfactual outcome estimation, which is employed to manipulate the model's cognition, treating poisoned samples as clean.

## 4.1 TRAINING STAGE: REMOVE BACKDOOR FROM CAUSAL VIEW

Here we introduce some basic definitions in causal inference:

**Definition 4.1** ($do$-Calculus). The operator $do(i')$ is defined in do-calculus as the intervention that sets the value of a variable $i$ to a constant $i'$ uniformly across the entire population.

**Definition 4.2** (Backdoor Path). A path connecting image $I$ and outcome $Y$ is defined as a backdoor path if and only if it is not a directed path and remains unblocked by any other variables.

**Definition 4.3** (Backdoor Criterion). A set of variables $A$ fulfills the backdoor criterion relative to $I$ and $Y$ if $A$ blocks all backdoor paths from $I$ to $Y$ and does not contain any descendants of $I$.

To mitigate confounding bias brought by $A$, we employ $do$-**calculus** in causal inference to cut off the spurious path as shown in Fig. 2(b). Specifically, $do$-calculus conducts intervention by substituting the value of $I$ with a specific value, thereby cutting off the relationship between the variable $I$ and its parent $A$, effectively eliminating any incoming edges to $I$. Consequently, all associations between $I$ and $Y$ flow along the directed causal path, yielding backdoor-free predictions $\mathbb{E}(Y|I = do(i))$.

However, $\mathbb{E}(Y|I = do(i))$ is a causal estimand that is intractable to identify with the current dataset (Pearl, 2009). To address this, we employ the backdoor adjustment for causal identification, a theory that identifies outcomes without the influence of confounders (attacks) (Pearl, 2009), thereby recovering backdoored samples while maintaining high accuracy. Specifically, backdoor adjustment involves adjustments on variables that satisfy the backdoor criterion (Pearl, 2009). According to Definition 4.3, attack $A$ in our scenario is a sufficient adjustment set, since there is only a single backdoor path through $A$, adjusting for this single node $A$ is sufficient to block all backdoor paths. Hence, the backdoor adjustment theorem tailored for backdoor attacks is as follows:

**Theorem 4.4** (Backdoor Adjustment for Backdoor Attack). *Given the satisfaction of the backdoor criterion by $A$, the positivity assumption (Assumption 3.2), and the modularity assumption (Assumption 3.3), we can identify the causal effect of $I$ on $Y$.*

$$\mathbb{E}[Y|I = do(i)] = \mathbb{E}[\sum_a \mathbb{P}(Y|i,a)\mathbb{P}(a)] = \mathbb{E}_A \mathbb{E}\left[Y \middle| \underset{\text{Image Input}}{i}, \underset{\text{Attack}}{A}\right]. \tag{2}$$

*Proof.* We provide a detailed proof in Appendix C.2.

The theorem implies that clean predictions can be derived from a poisoned dataset by conditioning on both the input image $i$ and the attack $A$. In other words, the model input should include both the image and the associated attack indicator. This distinction also sheds light on the differences between training a backdoored model and a clean model on a poisoned dataset. Specifically, *simply inputting the image causes the model to be backdoored, while including both the image and the associated attack indicator can safeguard the integrity of the victim model*. Furthermore, the difference between equation 1 and equation 2 provides theoretical insight into why simply inputting the image causes the model to be backdoored. Specifically, we find that the biased estimation in equation 1 contains an additional term (highlighted in gray) compared to the backdoor-free estimation in equation 2. This additional term is the source of confounding bias (Pearl, 2009; Guo et al., 2020).

**How to represent the Attack $A$?** An intuitive way for indicating the attack is leveraging off-the-shelf backdoor detection algorithms (Koh & Liang, 2017; Peri et al., 2020; Qi et al., 2023; Pan et al., 2023). However, their reliance on already backdoored models and computationally intensive requirements make these methods impractical for representing $A$ as an input in our approach. Inspired by the input-level detection method (Zeng et al., 2021), we use the frequency spectrum of each image to indicate the presence of an attack on the corresponding image. This method is effective due to the significant differences between backdoored and clean images in the frequency domain, as illustrated in Fig.1. Its effectiveness, along with its proactive and computationally efficient nature, which does not depend on an already trained backdoored model, makes it a highly appropriate choice of an attack. More visualization can be found in Fig. 14. Moreover, images can be efficiently transformed into the frequency domain using the Type-II 2D Discrete Cosine Transform (DCT). Consequently, we incorporate a structure that accepts DCT-transformed images as input and maps them to a vector representing the attack. Further details on using frequency domain images to detect backdoors are discussed in Appendix E.

**MCCI Design.** Motivated by our previous causal analysis, we propose our Mind Control through Causal Inference model (MCCI), which consists of two networks: the Semantic Feature Recognition

Network (SFRN) and the Attack Indication Network (AIN) as shown in Fig. 1. Specifically, SFRN is the provided victim model, such as ViT. It is tasked with recognizing complex semantic features, such as identifying a cat within an image, thereby functioning as the clean model that learns direct causal paths. This process, illustrated in the SFRN latent space in Fig.1, involves processing the original image $x$ and encoding it into $z$. Meanwhile, AIN determines whether the input is back-doored based on the frequency spectrum $x_f$ of $x$ as illustrated in the AIN latent space in Fig. 1. It acts like a magic wand, **controlling the model's perception** to recognize whether the input is clean or backdoored. The frequency spectrum $x_f$ is encoded by AIN into $z_f$. Then two embeddings ($z$ and $z_f$) are concatenated for the final image prediction. More detail can be found in D.

*Remark* 4.5 (Design Guidelines for AIN). We propose that an AIN module should satisfy the following two requirements: (1) The structure is much smaller than SFRN, as backdoor patterns are simpler to learn than normal patterns, as evidenced by (Liu et al., 2023b; Zhang et al., 2023); (2) Avoids using complex architectures to decrease unnecessary overhead. Further discussion on AIN's "magic control power" is in Section 5.2.

In summary, the training pipeline of MCCI becomes solving the following optimization problem:

$$\theta_{AIN}^*, \theta_{SFRN}^* = \underset{\theta_{AIN}, \theta_{SFRN}}{\arg\min} \; \mathbb{E}_{(x,y)\sim\mathcal{D}_{train}}\ell(C(g(z, z_f)), y)$$
$$\text{where} \quad z = AIN_{\theta_{AIN}}(DCT(x)), z_f = SFRN_{\theta_{SFRN}}(x), \tag{3}$$

where $\ell$ denotes the cross-entropy loss function, $C$ denotes a composited classification layer, $g$ denotes the concatenation function, and $DCT$ denotes the discrete cosine transform function. The outline of the algorithm is provided in Algorithm 1.

## 4.2 INFERENCE STAGE: ASK MODEL WHAT IF IT WERE A CLEAN IMAGE

In the inference stage, we use counterfactual outcome estimation to predict the original ground-truth label for each image. This method estimates what the outcome would be if the variable had a different value than the observed one in the factual world. Specifically, for a poisoned image $i$, we aim to determine what its label would be if it were a clean image. Answering this counterfactual question involves estimating $\mathbb{E}[Y|I = do(i), A = 0]$. This motivates us to input a fake non-attack indicator, namely, a clean frequency spectrum ($A = 0$) to trigger the control magic of AIN for prediction. Intuitively, the clean frequency spectrum and the trained AIN work together to control the model's perception, convincing it that the input is clean and thereby guiding it to the correct prediction. Moreover, we discuss this "magic control secret" further in Section 5.2. The prediction pipeline can be formulated as follows,

$$\hat{y} = C(g(AIN_{\theta_{AIN}^*}(DCT_{benign}, SFRN_{\theta_{SFRN}^*}(x)))), \tag{4}$$

where $DCT_{benign}$ denotes the clean frequency spectrum. Regarding the choice of the clean frequency spectrum, we propose two heuristic strategies: (1) the average frequency spectrum derived from a number of clean images in the validation dataset, (2) the frequency spectrum of a single clean image chosen randomly from the validation dataset. Our experiments show that both of the methods work, namely, only one clean image is needed for satisfactory defense, demonstrating the practicality of our method. More details can be found in Section 5.2.

## 5 EXPERIMENTS

**Datasets and Models.** Following (Guo et al., 2023a; Gao et al., 2019; Li et al., 2021a), we choose two widely-adopted datasets for evaluating the effectiveness of our proposed method: CIFAR-10 (Krizhevsky, 2009), and ImageNet-subset (Deng et al., 2009). The details of the datasets are listed in Appendix F. For CIFAR-10, we train with the widely-adopted ResNet-18 (He et al., 2015). For the ImageNet-subset, we opted for the EfficientNet architecture (Tan & Le, 2020) as it reports a higher accuracy.

**Attack & Defense Baselines.** We choose six backdoor attacks from the well-established recent works as our baselines: 1) BadNet (Gu et al., 2017), 2) Blend Attack (Chen et al., 2017), 3) Label-Clean backdoor attacks (Shafahi et al., 2018b), 4) WaNet (Nguyen & Tran, 2021), 5) ISSBA (Li et al., 2021c), and 6) TUAP (Zhang et al., 2021). All attack baselines are implemented with the

Table 1: Comparison of the proposed method with other baseline defense methods. We mark the best values in **bold** and the second best values in underline.

| Dataset | Attack Method ↓ | No Defense | | ABL | | | CBD | | | DBD | | | ASD | | | MCCI-AVG | | | MCCI-SG | | |
|---|---|---|---|---|---|---|---|---|---|---|---|---|---|---|---|---|---|---|---|---|---|
| | | CA ↑ | ASR ↓ | CA ↑ | ASR ↓ | ARR ↑ | CA ↑ | ASR ↓ | ARR ↑ | CA ↑ | ASR ↓ | ARR ↑ | CA ↑ | ASR ↓ | ARR ↑ | CA ↑ | ASR ↓ | ARR ↑ | CA ↑ | ASR ↓ | ARR ↑ |
| CIFAR-10 | BadNet | 87.22 | 100.00 | 81.64 | 5.73 | 81.88 | 76.31 | 2.44 | 75.17 | 86.67 | 1.52 | 83.49 | 86.17 | 0.35 | **85.91** | **86.73** | 0.50 | 85.64 | 86.17 | 2.62 | 85.08 |
| | Blend | 87.55 | 100.00 | 82.87 | 1.69 | 77.01 | 77.18 | 97.04 | 12.27 | 85.54 | 2.70 | **86.86** | 83.94 | 2.21 | 82.69 | **85.83** | 1.94 | 80.05 | 85.23 | **1.07** | 78.18 |
| | WaNet | 86.89 | 99.70 | 83.33 | 0.48 | 84.63 | 75.82 | 60.27 | 36.78 | 84.43 | 4.10 | 77.17 | 84.67 | **2.72** | 83.72 | 81.04 | 15.54 | 75.98 | **85.56** | 7.75 | **80.19** |
| | ISSBA | 87.80 | 99.42 | 85.84 | 0.94 | 85.03 | 76.64 | 13.99 | 15.50 | 77.27 | 14.90 | 78.43 | 76.79 | 4.16 | 12.60 | 78.19 | **1.02** | 78.93 | **79.22** | 3.38 | **79.71** |
| | LC | 86.81 | 68.21 | 68.97 | 27.69 | 63.58 | 78.37 | 6.42 | 72.32 | 85.02 | 3.33 | 63.01 | 85.50 | 1.14 | **85.41** | **85.51** | 0.48 | 81.43 | 85.50 | 1.23 | 79.62 |
| | TUAP | 87.35 | 87.66 | 81.12 | 5.26 | 81.18 | 76.08 | 2.76 | 75.71 | 86.26 | 1.85 | 83.04 | 86.47 | 2.03 | 28.26 | **86.56** | **0.73** | 85.18 | 86.53 | 2.54 | **85.40** |
| | **Average** | 87.27 | 92.50 | 80.63 | 6.97 | 78.89 | 76.73 | 30.49 | 47.96 | **84.70** | 4.73 | 78.67 | 83.92 | **2.10** | 63.10 | 83.98 | 3.37 | 81.20 | **84.70** | 3.10 | **81.36** |
| ImageNet-subset | BadNet | 72.11 | 100.00 | 70.52 | 12.24 | 72.92 | 57.19 | 26.89 | 50.48 | **70.56** | 3.51 | **75.60** | 70.03 | 3.42 | 68.56 | 70.51 | 0.48 | 70.44 | 70.24 | 1.82 | 70.67 |
| | Blend | 72.24 | 100.00 | 67.31 | 53.82 | 43.76 | 49.70 | 27.41 | 43.07 | 70.80 | 4.72 | 66.27 | 70.22 | 9.36 | 67.33 | **70.89** | 2.58 | **66.94** | 70.20 | 3.32 | 66.54 |
| | WaNet | 70.44 | 98.66 | 69.33 | 97.31 | 11.97 | 65.40 | 99.84 | 10.65 | 69.30 | 5.97 | 63.30 | 69.12 | 10.12 | 58.33 | 69.72 | **0.06** | 60.20 | **70.21** | 0.51 | **69.56** |
| | ISSBA | 71.55 | 93.55 | 67.76 | 43.58 | 21.77 | 44.89 | 100.00 | 9.45 | 65.74 | 10.90 | **57.74** | 68.97 | 9.63 | 13.26 | 70.77 | **0.01** | 28.16 | **70.00** | 0.76 | 38.32 |
| | LC | 72.27 | 72.41 | 66.74 | 0.81 | 65.08 | 61.76 | 11.19 | 61.38 | 69.46 | 3.44 | 47.96 | 69.23 | 2.11 | 53.21 | **69.96** | 0.43 | 65.26 | 69.56 | **0.14** | **67.99** |
| | TUAP | 70.79 | 79.02 | 64.86 | 14.60 | 43.32 | 56.41 | 23.69 | 42.33 | **69.29** | **2.68** | 43.36 | 68.12 | 8.95 | 32.15 | 69.23 | 10.16 | 60.65 | 69.22 | 8.87 | **65.01** |
| | **Average** | 71.57 | 90.61 | 67.75 | 37.06 | 43.14 | 55.89 | 48.17 | 36.23 | 69.19 | 5.20 | 59.04 | 62.62 | 7.27 | 48.81 | **70.18** | 2.31 | 65.28 | 69.91 | **2.60** | **67.33** |

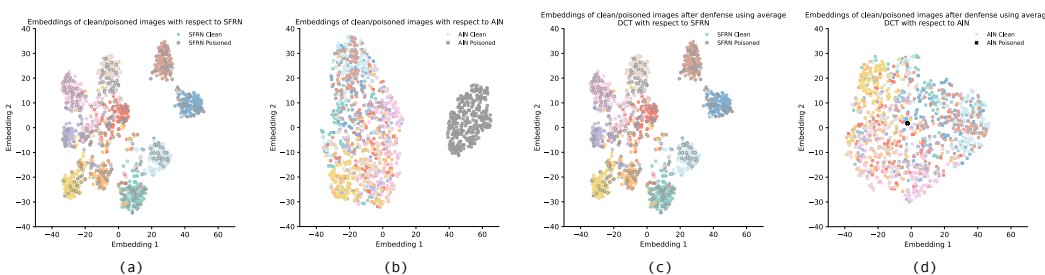

Figure 3: t-SNE visualisation of embeddings after Semantic Feature Recognition Network (SFRN) and the Attack Indication Network (AIN) in terms of clean and poisoned points under the training stage ((a)-(b)) and the inference stage using average clean frequency spectrum ((c)-(d)).

open-sourced backdoor learning toolbox (Li et al., 2023). More details of each attack method can be found in Appendix G. Based on our setting, it is assumed that defenders can only access the poisoned dataset and have full control over the training process but do not have prior knowledge about the poisoned distribution as well as the trigger pattern. Hence we adopt ABL (Li et al., 2021a), CBD (Zhang et al., 2023), DBD (Huang et al., 2022), and ASD (Gao et al., 2023) for the comparision. More details about defense baselines can be found in Appendix H.

**Evaluation Metrics.** Following existing works in backdoor suppression (Gao et al., 2021; Guo et al., 2021; 2023a), we use Clean Accuracy (CA) and Attack Success Rate (ASR) as evaluation metrics. In addition, we also adopt **Attack Recovery Rate (ARR)**, which measures the proportion of recovered backdoored samples in the poisoned dataset. More details can be found in Appendix L.

**Implementation Details.** The full implementation details are provided in Appendix K.

## 5.1 MAIN RESULTS

### 5.1.1 DEFENSE EFFECTIVENESS

Table 1 compares our method MCCI with other defense baselines across various backdoor attacks on two datasets. We highlight the best results in **bold** and the second best in underline. MCCI-AVG represents our defense model using an averaged frequency spectrum from a small number of clean images, while MCCI-SG uses a frequency spectrum from a randomly selected clean image. The table shows that our methods achieve promising performance across all datasets against various attack methods, not only on conventional evaluation metrics like CA and ASR but also on the newly adopted ARR, which assesses the recovery of poisoned samples to their original correct labels. In particular, the ARR of our models is nearly equivalent to CA, indicating that our methods can effectively recover all poisoned images as long as their clean counterparts are correctly classified by the model. However, baseline models perform poorly on the ARR metric, in line with previous studies that suggest achieving a high ARR is challenging, even for models with low ASRs (TRC'22, 2022; Wu et al., 2022). There is no significant difference between MCCI-AVG and MCCI-SG, suggesting that *one randomly selected clean image* is sufficient for defense. This highlights the

practicality of our model, as only one clean image is required for defense, which is easily obtainable by the defender.

### 5.1.2 ADAPTATION TO LARGE PRE-TRAINED MODELS

MCCI could also be easily adapted to the trending large pre-trained models such as ViT (Dosovit-skiy, 2020), CLIP (Radford et al., 2021), and BLIP (Li et al., 2022a). In this section, we manually inject BadNet-style backdoors into these three pre-trained models during the finetuning stage. Then we evaluate whether MCCI could still recover the original correct targets when provided with a clean frequency spectrum. Due to space limitations, the attack details for the three foundation models are provided in the Appendix M.

For the ViT and CLIP models, we consider fine-tuning them on CIFAR-10 and Fashion-Products datasets, respectively. We provide the results in Table 2. As observed, the ASR values drops abruptly when provided with a clean frequency spectrum, and the ARR rate maintains high. This demonstrates that MCCI performs well even when integrated with large pre-trained models in the classification task. We also fine-tune the BLIP model for a VQA task on the poisoned IconDomain dataset. It is noted that we slightly abuse the notation of CA, ASR, and ARR here (more details are in the Appendix M). As observed in Table 2, the CA and ARR are closed to zero, demonstrating that MCCI could effectively recover the original QA capability when provided with clean frequency spectrum. Meanwhile, the ASR values are significantly higher, showing that the model's strong effectiveness in refusing backdoor target answers.

Table 2: Effectiveness of MCCI on large pre-train models.

| Dataset | Backbone Model | No Defense | | MCCI-AVG | | | MCCI-SG | | |
|---|---|---|---|---|---|---|---|---|---|
| | | CA | ASR | CA | ASR | ARR | CA | ASR | ARR |
| CIFAR-10 | ViT | 96.45 | 97.00 | 84.21 | 0.00 | 75.31 | 97.34 | 0.00 | 96.39 |
| Fashion-Products | CLIP | 96.20 | 99.72 | 95.83 | 2.72 | 92.02 | 95.47 | 3.02 | 91.24 |
| Icon-Domain | BLIP [*] | 0.12 | 0.00 | 0.13 | 3.21 | 0.38 | 0.13 | 3.71 | 0.25 |

[*] The metrics are slightly abused for the VQA experiments with BLIP.

### 5.2 ABLATION STUDY

**Mind Control Effectiveness of AIN.** One of the key contributions of our work is the AIN's "magic control power" over the model's perception. We explore whether it enables control over the model's recognition of inputs as either clean or poisoned based on the frequency spectrum of images. Previous results in Table 1 have shown that introducing a clean frequency image effectively causes the model to misidentify a poisoned image as clean, while minimally impacting the correct prediction of clean images. Building on this, we now explore the opposite scenario: whether the model can be misled

Table 3: The performance of the mind control magic.

| Attack Method ↓ | APC | APP | RPC | RPP |
|---|---|---|---|---|
| BadNet | 10.27 | 100.00 | 10.03 | 100.00 |
| Blend | 10.35 | 100.00 | 10.46 | 100.00 |
| WaNet | 13.27 | 97.54 | 10.63 | 100.00 |
| ISSBA | 10.04 | 100.00 | 10.32 | 100.00 |
| LC | 19.67 | 94.02 | 21.09 | 92.78 |
| TUAP | 61.84 | 64.47 | 39.92 | 85.62 |

into recognizing a clean image as poisoned using the average frequency spectrum of poisoned data or that of a randomly chosen poisoned image. Specifically, we conduct experiments on CIFAR-10, with results shown in Table 3. Here, APC and APP represent the accuracy of using the average poisoned frequency spectrum (AP) on clean and poisoned datasets, respectively. The near $10\%$ APC suggests that AP leads the model to recognize all clean images as poisoned, aligning with the fact that the ground truth for $10\%$ of clean images is the target label. The near $100\%$ APP indicates that AP enhances the model's tendency to classify poisoned images as such, improving the ASR. Similar results were observed with the random poisoned frequency spectrum (RPC and RPP).

**Reveal the magic secret of SFRN and AIN.** To better understand MCCI and its components, we present t-SNE visualizations of embeddings from SFRN (round points) and AIN (cross points) under the BadNet attack on CIFAR-10 in Fig. 3. Clean images are represented in various colors, each corresponding to a different class, while poisoned images are circled in gray, with the interior color representing the original ground-truth label. From left to right, we present the embeddings for SFRN and AIN under the training stage ((a)-(b)), and the inference stage ((c)-(d)). Fig. 3(a) shows that, the input images are mapped to different clusters by SFRN, demonstrating that in the training stage, the SFRN module mainly captures the semantic information of the images. Fig. 3(b) shows that, the embeddings of the AIN module are split into two disjoint clusters, where the smaller one contains the embeddings for the backdoored images, and the larger one contains the embeddings for the clean images. This demonstrates that AIN is capable of distinguishing whether the image is backdoored or

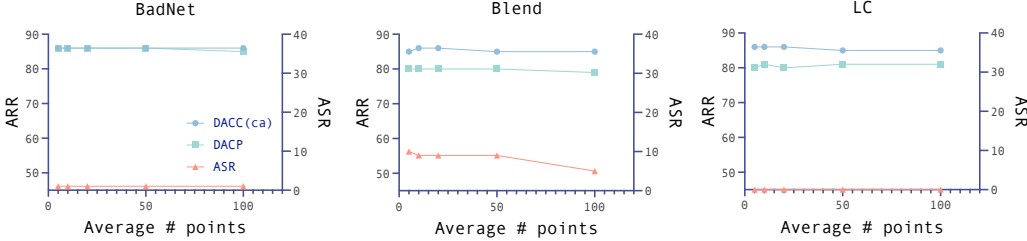

Figure 4: Sensitivity analysis with different quantities of clean images used to calculate the average frequency spectrum on CIFAR-10 datasets.

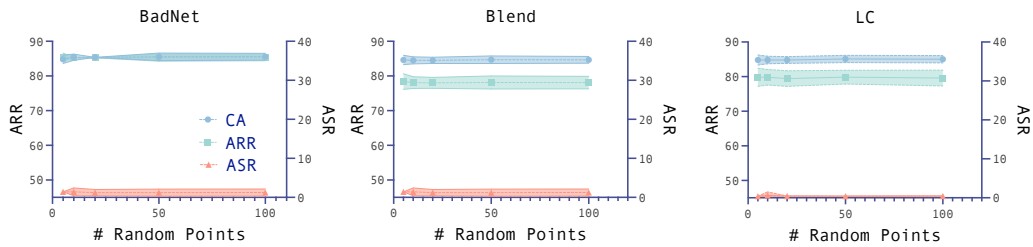

Figure 5: The mean (line) and standard deviation (band) of the performance under different randomly selected points.

not, but not able to recognize complex semantic information in the input images. These observations further confirm that the two networks function as expected, as discussed in Section 4.1.

Fig. 3 (c) and (d) reveal the magic secret of controlling the model to recognize a poisoned sample as a clean sample. For AIN, using average clean DCT shifts the embeddings of poisoned images into the clean image group. Surprisingly, we find that SFRN already groups poisoned images into their original ground-truth categories even before the defense. This occurs because SFRN prioritizes the main semantic features in the image, disregarding the semantically meaningless trigger patterns. As a result, it clusters poisoned images based on their primary features into the correct classes. Based on this insight, we design a new structure that utilizes only SFRN for defense. Details can be found in Appendix O. Thus, after the defense, both AIN and SFRN embeddings of poisoned images align with those of clean images, which explains how the model restores the ground-truth labels of poisoned images. Randomly selected DCT shows the same pattern as detailed in the Appendix N.

**Choice of Clean Frequency Spectrum.** We examine the impact of different selections of the clean-frequency spectrum on defense performance against different attacks on CIFAR-10. Detailed results for other attacks can be found in Appendix J. For the average frequency spectrum, we investigate how the number of clean images used to calculate the spectrum affects the model performance. Results in Fig. 4 show that *even a limited set of clean images (e.g., 5) is sufficient for a successful defense*, with minimal performance improvement as the number of clean images increases. Regarding the random frequency spectrum, we assess whether the frequency spectrum of any clean image can be used for inference. We randomly select subsets of $[5, 10, 20, 50, 100]$ images and calculate the average and standard deviation of their performance. Fig. 5 shows that the average performance remains relatively consistent regardless of the number of images used, and the standard deviation is small, suggesting that *no carefully selected clean image is necessary for effective inference*.

## 5.3 DISCUSSION

**Effectiveness with Different Poisoning Rates.** We study the robustness of our method against different poisoning rates. Specifically, we test our method with poisoning rates of $[5\%, 10\%, 15\%, 20\%, 50\%, 70\%]$. The results, shown in Fig. 7, demonstrate that our MCCI model maintains a high CA, ARR, and a low ASR across a wide range of poisoning rates. Specifically, the

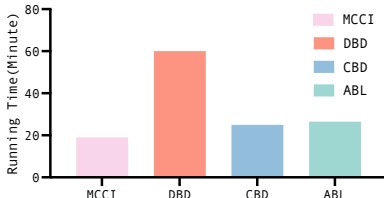

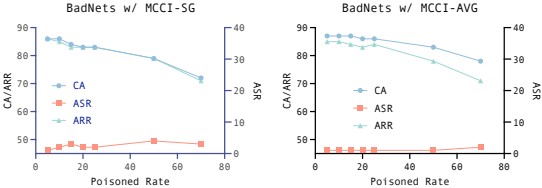

Figure 6: Efficiency comparison.

Figure 7: Robustness test with the poisoning rate from $5\%$ to $70\%$ for BadNets on CIFAR-10.

average results are $0.82$ CA, $0.01$ ASR, and $0.81$ ARR for AVG, and $0.82$ CA, $0.02$ ASR, and $0.81$ ARR for SG, indicating that our MCCI is robust across varying poisoning rates.

**Efficiency Test.** We compare the efficiency of training a clean model on the poisoned CIFAR-10 dataset with current SOTA methods. Specifically, we record the running time that the trained models achieve a clean accuracy higher than 85% on CIFAR-10 dataset. Results in Fig. 6 show that MCCI requires $24\% - 215\%$ less training time than baselines, indicating the high efficiency of MCCI.

**Resistance to Potential Adaptive Attack.** While it was not our initial intention, it is conceivable that adversaries could fully understand our defense method, leading to the development of more sophisticated backdoor attacks designed to bypass our defenses. Specifically, adversaries might aim to modify the frequency spectrum of poisoned images so that they contain only low-frequency components, similar to those of clean images, while keeping the clean images in the training set unchanged to maintain stealth. Formally, the objective can be expressed as

Table 4: The performance of MCCI against different adaptive attacks

| Techniques | No Defense | | MCCI-AVG | | | MCCI-SG | | |
|---|---|---|---|---|---|---|---|---|
| | CA↑ | ASR↓ | CA↑ | ASR↓ | ARR↑ | CA↑ | ASR↓ | ARR↑ |
| Averaging | 87.20 | 100.00 | 84.65 | 3.32 | 64.05 | 85.98 | 3.22 | 63.20 |
| Gaussian | 87.51 | 100.00 | 84.65 | 5.12 | 70.06 | 85.51 | 3.25 | 70.98 |
| Median | 87.05 | 100.00 | 85.37 | 1.64 | 75.86 | 85.27 | 2.93 | 76.41 |

$\min_\theta \sum_{i=1}^{|\mathcal{D}_b|} \ell(f_\theta(\hat{\boldsymbol{x}}_i), y_t) + \sum_{i=1}^{|\mathcal{D}_c|} \ell(f_\theta(\boldsymbol{x}_i), y_i)$, where $f_\theta$ represents the victim model, $D_b$ and $D_c$ are the poisoned and clean sets in the training set, respectively. $\hat{\boldsymbol{x}}_i, y_t$ represent poisoned images, which contain only low-frequency components and their target labels, while $\boldsymbol{x}_i, y_i$ represent clean images and their corresponding labels. Specifically, we employ three widely adopted low-pass filter kernels: Averaging Blur, Gaussian Blur, and Median Blur to obtain smoother poisoned images $\hat{\boldsymbol{x}}_i$. We adopt BadNets on the CIFAR-10 dataset as an example for our discussions. The results shown in Table 4 indicate that our model still achieves promising performance after applying these filters, with ASR lower than $0.05$ and ARR around $0.70$ among all adaptive attacks. This demonstrates our model's robustness to adaptive attacks. More details are in Appendix P and Q.

## 6 CONCLUSION AND FUTURE WORKS

In this paper, we focus on the anti-backdoor learning problem with a practical objective: *Recovering the original, correct predictions for the backdoored images*. We first analyze the problem from a novel causal view, deriving that backdoor attacks work as a confounder, opening a spurious path that builds and strengthens erroneous correlations between the poisoned images and the target label. To mitigate the confounder, we conduct backdoor adjustment by introducing Mind Control through Causal Inference (MCCI), which efficiently models the causal effect from training images with a semantic feature recognition network (SFRN) and an Attack Indication Network (AIN). The extensive experiments across various datasets demonstrate that our method can not only avoid misclassifying backdoor images, but also recover the original, correct predictions for the backdoor images. Despite the promising results, several interesting future directions remain: (1) Using the frequency spectrum to represent attacks relies on the assumption that backdoor triggers contain high-frequency components. Are there more general and intuitive methods to represent the attack? Are there any efficient detection algorithms that could be used as AIN? (2) How to design stronger attacks that circumvent our defense mechanism, and (3) Adapting the MCCI to the NLP domain is also interesting. More details can be found in R.

## 7 ACKNOWLEDGEMENTS

This work is supported in part through grants from the Amazon-Virginia Tech Initiative for Efficient and Robust Machine Learning, the National Science Foundation under Grant numbers IIS-2316306, CNS-2330215, CNS-2424127, CCF-1918656, CNS-2317193 and IIS-2331315, the U.S. Office of Naval Research Award under Grant Number N00014-24-1-2668, the Cisco Award, the Commonwealth Cyber Initiative Cybersecurity Research Award, and the VT 4-VA Collaborative Research Grant.

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

Figure 8: Visualization of the Threat Model.

## A  VISUALIZATION OF THE THREAT MODEL

In this paper, the defender aims to train a clean model from a poisoned dataset with the following two objectives: ❶ the clean model is expected to produce high-accuracy predictions for the clean samples, achieving performance as good as the model trained on purely clean dataset, and ❷ the clean model is expected to produce the original, correct predictions for the backdoored images even if the trigger is present. We assume that the defender has full control of the training process but lacks prior knowledge about the proportion and the distribution of backdoor samples in the poisoned dataset. Fig. 8 offers a visualization of the considered threat model.

## B  CAUSAL ASSUMPTIONS

Here we present the basic definitions, assumptions, and corresponding explanations in causal inference.

**Assumption B.1** (Stable Unit Treatment Value Assumption (SUTVA)). There are no interactions between units, and each treatment has only one version. Different levels or doses of a treatment are considered as different treatments.

SUTVA comprises two conditions: well-defined treatment levels and no interference (Guo et al., 2020). The well-defined treatment condition means that if two different instances $i$ and $j$ have the same value for their treatment variable, they receive the same treatment. The no interference condition signifies that the potential outcomes of an instance are independent of the treatments received by other units.

**Assumption B.2** (Ignorability). The potential outcome $Y(T = t)$ is independent of the treatment assignment given all covariates. Formally, $Y(T = t) \perp\!\!\!\perp t|X$.

Ignorability ensures that the outcome is not affected by the assignment of treatment when condition on $X$, making the treatment group and control group comparable after conditioning on $X$.

**Assumption B.3** (Positivity). Every unit should have non-zero probabilities to be assigned in each treatment group. Formally, $\mathbb{P}(T = t|X = x) \neq 0, \forall t \in \mathcal{T}, \forall x \in X$.

Positivity is the condition that all subgroups of the data with different covariates have a nonzero probability of receiving any treatment value. In short, a positivity violation means that we are conditioning on an event with zero probability.

**Assumption B.4** (Modularity). Given a node $x_i$ and its parent nodes $pa_i$ in the causal graph. If we intervene on a set of nodes $S$ by assigning them to constants, then for all $i$: If $i \notin S$, then $\mathbb{P}(x_i|pa_i)$ remains unchanged. If $i \in S$ and $x_i$ is the value that $X_i$ was set to by the intervention, then $\mathbb{P}(x_i|pa_i) = 1$; otherwise, $\mathbb{P}(x_i|pa_i) = 0$.

The modularity assumption ensures that an intervention is local. In other words, intervening on a variable $X$ changes only the causal mechanism for $X$ and does not alter the causal mechanisms that generate any other variables.

**Definition B.5** (Confounders). Confounders are variables that serve as a common cause of the treatment $I$ and the outcome $Y$.

## C  DEFINITIONS AND PROOFS

### C.1  MATH DEVIATION

**Proposition C.1.** *Let $i$ be an image from a poisoned dataset and $f_\theta$ a model trained on this dataset. Then, the expected prediction $Y$ of the model for image $I = i$ can be derived according to Fig. 2:*

$$\mathbb{E}[Y|I = i] = \mathbb{E}_A[\mathbb{E}\left[Y|i, A\right] \frac{P(i|A)}{P(i)}]. \tag{5}$$

*Proof.*

$$
\begin{aligned}
P(y|i) &= \sum_a P(y, a|i) \\
&= \sum_a P(y|i, a)P(a|i) \\
&= \sum_a P(y|i, a)\frac{P(i|a)}{P(i)}P(a) \\
&= \mathbb{E}_A[P(y|i, A)\frac{P(i|A)}{P(i)}].
\end{aligned}
\tag{6}
$$

The first equality is by the definition of marginal distribution. The second and third equalities are by the Bayes' theorem. The fourth equality is by the definition of expectation.

Then we take an expectation over $Y$, we have:

$$\mathbb{E}[Y|I = i] = \mathbb{E}[\mathbb{E}_A[P(Y|i, A)\frac{P(i|A)}{P(i)}]] = \mathbb{E}_A[\mathbb{E}\left[Y|i, A\right] \frac{P(i|A)}{P(i)}]. \tag{7}$$

The second equality is defined by the expectation. Specifically, $Y$ is a vector of outcomes, where each element in $Y$ represents the probability of a specific class.

### C.2  PROOF OF BACKDOOR ADJUSTMENT

**Theorem C.2** (Backdoor Adjustment for Backdoor Attack)**.** *Given the satisfaction of the backdoor criterion by $A$ (Definition 4.3, and the positivity assumption 3.2, we can identify the causal effect of $I$ on $Y$.*

$$\mathbb{E}[Y|I = do(i)] = \mathbb{E}[\sum_a \mathbb{P}(Y|i, a)\mathbb{P}(a)] = \mathbb{E}_A\mathbb{E}\left[Y|i, A\right]. \tag{8}$$

*Proof.*

$$
\begin{aligned}
\mathbb{P}(y|do(i)) &= \sum_a \mathbb{P}(y|do(i), a)\mathbb{P}(a|do(i)) \\
&= \sum_a \mathbb{P}(y|i, a)\mathbb{P}(a|do(i)) \\
&= \sum_a \mathbb{P}(y|i, a)\mathbb{P}(a).
\end{aligned}
\tag{9}
$$

The first equality is by the law of total probability. The second equality follows the modularity assumption 3.3. Since $A$ are all the parents for $Y$ in addition to $I$, hence $\mathbb{P}(y|do(i), a) = \mathbb{P}(y|i, a)$. The third equality is obtained by the do-calculas, since all the incoming edges for $I$ have been cut off, there is no relationship between $I$ and $A$.

Then we take an expectation over $Y$, we have:

$$\mathbb{E}[Y|I = do(i)] = \mathbb{E}[\sum_a \mathbb{P}(Y|i, a)\mathbb{P}(a)] = \mathbb{E}_A\mathbb{E}\left[Y|i, A\right]. \tag{10}$$

The second equality is by the definition of the expectation. Specifically, $Y$ is a vector of outcomes, where each element in $Y$ represents the probability of a specific class.

$\square$

## D    More Details About the Design and Function of MCCI

Inspired by the theorem 4.4, we concluded that achieving an unbiased output requires inputting both the image $i$ and the attack indicator $A$ into the model, for which we use the frequency spectrum as the indicator. While using the image as input is straightforward, as most models already process images, incorporating the additional attack indicator presents a challenge. To address this, we designed an additional network, AIN, to process the attack indicator $A$ while leaving the original victim model (SFRN, e.g. ViT) unchanged. The input to the SFRN remains the image, and the encoded outputs from both networks are concatenated in the final layers for prediction. As shown in $\mathbb{E}\left[Y|i, A\right]$, making the final prediction requires two pieces of information: the information from both the attack indicator and the image. The AIN processes $A$ and provides the encoded attack information, while the SFRN processes the image and provides the encoded image information. Both are combined before making the final prediction. Thus, the construction of SFRN and AIN is primarily inspired by the theoretical framework of $\mathbb{E}\left[Y|i, A\right]$.

From an empirical perspective, as shown in Fig. 3, the results further validate our method. The AIN effectively learns how to distinguish between clean and backdoored samples (successfully encodes the attack information), while the SFRN focuses on learning semantic information (successfully encodes the image information). This phenomenon might result in the high prominence of the AIN's predictions, which dominate the reduction of learning loss on poisoned samples, thus minimizing the SFRN's learning of the backdoor. Since AIN extracts the poisoned portion from the model, allowing the SFRN to primarily focus on learning semantic features rather than backdoored patterns. While the input to the SFRN may still contain trigger patterns, these patterns fail to activate the backdoor because the SFRN interprets them as noise rather than meaningful triggers of the attack (effectively cutting off $A \rightarrow I$), which aligns with the motivation mentioned in Section 3.3.

## E    More Details About the Frequency Domain

In essence, the term 'frequency' in image processing refers to the rate at which pixel values change. High-frequency components are typically indicative of edges within an image, where there are abrupt changes in pixel values. Conversely, low-frequency components are associated with smoother regions in an image, where changes in pixel values are more gradual. Typically, the frequency spectrum is widely used to visualize the image in the frequency domain. Previous studies show that low-frequency components dominate the frequency spectrum of the nature image, since colors always change gradually in images and sudden changes in pixel values (e.g., edges in images) are relatively scarce (Zeng et al., 2021; Burton & Moorhead, 1987; Tolhurst et al., 1992). However, for backdoored images, the specific triggers make the picture less smooth and always result in high-frequency artifact components, since they either decrease the correlation between neighboring pixels or the intrinsic high-frequency artifacts carried by them (Zeng et al., 2021). Hence, image frequency can be used as an indicator to indicate the presence of an attack on a single image $A$.

To convert images into the frequency domain, we apply the Type II 2D Discrete Cosine Transform (DCT), following the approach outlined in (Zeng et al., 2021). Similar to the Discrete Fourier Transform (DFT), the DCT interprets an image by representing it as a collection of cosine functions, each defined by distinct magnitudes and frequencies. This technique is widely recognized and used in various image compression algorithms, including JPEG.

## F    More Details About the Dataset

The details of the dataset are given in Table 5.

Table 5: Statistical information about the Datasets

| Dataset | Image Size | # of Training samples | # of Testing Samples | # of Classes |
|---|---|---|---|---|
| CIFAR-10 | $32 \times 32 \times 3$ | 50,000 | 10,000 | 10 |
| ImageNet-Subset | $224 \times 224 \times 3$ | 9,469 | 3,925 | 10 |

## G  MORE DETAILS ABOUT THE ATTACK BASELINES

All attack baselines are implemented with the open-sourced backdoor learning toolbox (Li et al., 2023). Below are the details of the these attack baselines:

- **BadNet** (Gu et al., 2017) employs 3×3 grid-like pixels as the triggers for each of the poisoned samples.
- **Blend** (Chen et al., 2017) uses a hello-kitty-like image and blends it with each of the poisoned samples.
- **WaNet** (Nguyen & Tran, 2021) employs the interpolation method and generates sample-specific triggers for each of the poisoned samples. We use noise ratio as two times of the poisoning ratio, grid size $k = 4$, grid_rescale=1, and warping strength $s = 0.5$.
- **ISSBA** (Li et al., 2021c) generates sample-specific trigger patterns through an encoder-decoder network. We employ the StegaStampEncoder model (https://github.com/tancik/StegaStamp), and train the model with only secret loss function for 2 epochs, and continue to train the model with total loss function for 18 epochs.
- **LC** (Shafahi et al., 2018b) proposes clean-label attacks, where the poisoned training data appear to be correctly labeled according to an expert observer by constructing adversarial samples. The parameters choice and implementations follow the well-established benchmarks BackdoorBox (Li et al., 2022b).
- **TUAP** (Zhang et al., 2021) proposes adding universal adversarial perturbations to victim images, causing them to move from their original classification region to a targeted region. We use $\epsilon = 0.031, \delta = 0.2$, overshoot=0.02, p_samples=0.01, and infinity norm as the norm function.

We present a visualization of the poisoned image generated by different backdoor attacks in Fig. 13.

## H  MORE DETAILS ABOUT THE DEFENSE BASELINES

A detailed descriptions of the chosen defense baselines are provided as follows:

**ABL (Li et al., 2021a)**  We follow the official implementation of ABL[1]. Specifically, ABL splits anti-backdoor training into three stages: isolation, finetuning, and unlearning. In the isolation stage, the model is trained by the local gradient ascent loss function with a few epochs (e.g., 20). Due to an interesting observation that models are easier to overfit on the backdoor samples than clean samples, the backdoor samples are isolated from the training dataset by picking samples with the top-k lowest loss values. In the finetuning stage, the model continues to train on the remaining training dataset. In the unlearning stage, the model unlearns the backdoors by using a naïve gradient ascent method over the isolated backdoor samples. For the hyperparameters in ABL, we follow all the default hyperparameters in the original implementations. We choose 20 as the tuning epochs, and 20 as the unlearning epochs. The $\gamma$ value in the local gradient ascent loss is chosen as 0.5.

**CBD (Zhang et al., 2023)**  We follow the official implementation of CBD[2]. Specifically, CBD splits anti-backdoor learning into two stages: First, a backdoored model is intentionally trained to capture the confounding effects (information about backdoor attacks). Then, in the next stage, CBD uses the other clean model to capture the desired causal effects by minimizing the mutual information with the confounding representations from the backdoored model and employs a sample-wise re-weighting scheme. We follow all the default hyperparameters in the original implementations. The DisenEstimator network used in the experiments is a WGAN-like network with a dropout rate of 0.2, and the backdoor training epoch is set as 5.

**DBD (Huang et al., 2022)**  We follow the implementation of public codes[3] on GitHub. Specifically, DBD splits anti-backdoor learning into three stages: self-supervised learning, supervised

---

[1]https://github.com/bboylyg/ABL

[2]https://github.com/zaixizhang/CBD

[3]https://github.com/SCLBD/DBD

learning, and semi-supervised learning. In the self-supervised learning stage, DBD trains an image encoder by conducting self-supervised learning over the training dataset (without labels) with an off-the-shelf SimCLR method. Then in the supervised learning stage, DBD freezes the parameters of the learned encoder and trains the remaining fully connected layers via standard training with all (labeled) training samples. In the third stage, to further sanitize the backdoors remaining in the model, DBD removes labels of some 'low-credible' samples determined based on the learned model and conducts a semi-supervised fine-tuning of the whole model. For the hyperparameters in DBD, we follow all the default hyperparameters in the original implementations. We choose 10 as the warm-up epochs for semi-supervised learning, and 100 as the warm-up epochs for self-supervised learning. $\epsilon$ is chosen as 0.5 by default.

**ASD (Gao et al., 2023)** ASD applies loss-guided split and meta-learning-inspired split to dynamically maintain and update two data pools. We follow all the default hyperparameters in the original implementations. We use Symmetric Cross Entropy loss as the splitting criterion, with $\alpha = 0.1, \beta = 1$. In the semi-supervised learning stage, we use mix-match loss with $\lambda_u = 15$ and train the model with $\alpha = 0.75$, temperature=0.5, and train iteration=1024.

## I   MORE DETAILS ABOUT THE RELATED WORK

Backdoor attacks are usually launched through data poisoning (Gu et al., 2017; Chen et al., 2017; Nguyen & Tran, 2021; Li et al., 2021c; Liu et al., 2020; Shafahi et al., 2018a; Schneider et al., 2024; Li et al., 2024b; Lan et al., 2024; Cheng et al., 2024; Yin et al., 2024), where malicious attackers inject backdoor samples into the training dataset. When a model is trained on this poisoned training dataset, a spurious correlation between the trigger and the target label is learned. In particular, different attack methods construct backdoor samples with different trigger patterns. For example, (Gu et al., 2017) adds a grid-style square trigger in the corner of clean images, (Chen et al., 2017) blends clean images with random pixels, (Nguyen & Tran, 2021) applies a warping operation to the original image, and (Liu et al., 2020) uses natural reflection to construct the backdoor trigger. In addition to data poisoning, there are many alternative ways to inject backdoors, such as supplying backdoored pre-trained models (Yao et al., 2019; Shen et al., 2021), tampering model weights and structures (Qi et al., 2021; Tang et al., 2020; Dong et al., 2023; Li et al., 2024a), and manipulating the training process (Li et al., 2021b; Doan et al., 2021). This paper primarily focuses on backdoor attacks through data poisoning.

## J   MORE DETAILS ABOUT THE CHOICE OF CLEAN FREQUENCY SPECTRUM

In this section, we demonstrate the performance under different choices of the clean frequency spectrum across other attacks in the main text. Specifically, Fig. 9 illustrates the performance across a different number of images used for obtaining the average frequency spectrum. The results indicate that averaging just 5 images is sufficient for satisfactory defense, and there is no significant improvement when increasing the number of images to average, which aligns with our findings in Section 5.2. The average and standard deviation of performance under different numbers of randomly selected images are shown in another Fig. 10. The high accuracy and low standard deviation demonstrate that any arbitrary clean image can be used for successful defense, highlighting the low requirements of our method.

## K   MORE DETAILS ABOUT THE IMPLEMENTATION

Following prior work in backdoor defenses (Li et al., 2021a), the poisoning ratio for all backdoor attacks is set to 10% by default. We use an initial learning rate of 0.1 that is decreased by a factor of 10 at epochs 30, 60, and 90, 100 epochs, a batch size of 128, and a weight decay of 1e-4 for training the defense model against all attack baselines. To conduct defenses on CIFAR-10, we use ResNet-18 for the SRFN structure. For the AIN structure, we adopt the same architecture as in (Zeng et al., 2021), as shown in Table 6. For AVG defense, we use 100 clean images for averaging by default. For SG defense, we randomly select one image from the clean dataset to conduct the defense. The poisoning rate is chosen as 10% by default, but we also evaluate the effectiveness of MCCI under different poisoning rates in Figure 7.

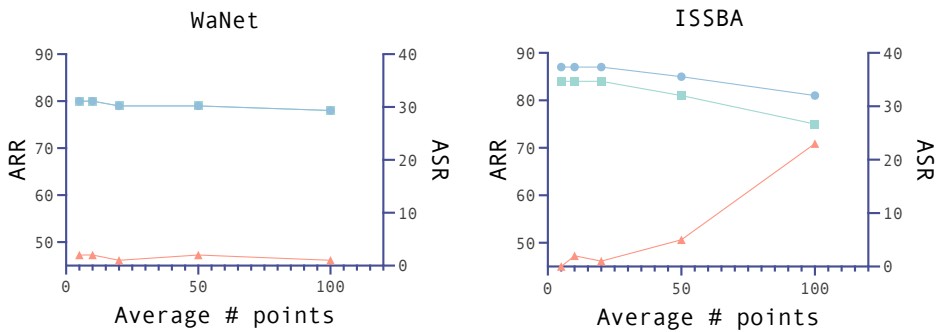

Figure 9: The mean (line) and standard deviation (band) of the performance under different randomly selected points.

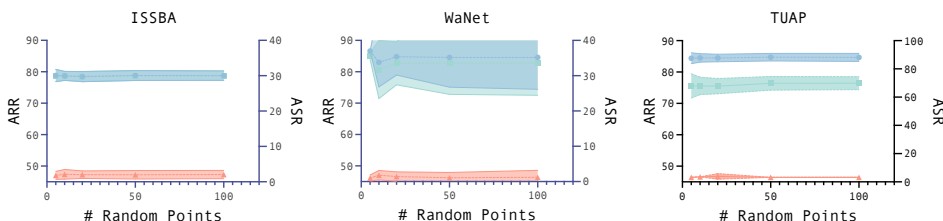

Figure 10: Sensitivity analysis with different quantities of clean images used to calculate the average frequency spectrum on CIFAR-10 datasets.

---

**Algorithm 1:** Mind Control Through Causal Inference (MCCI)

---

**Input:** Poisoned Dataset $\hat{\mathcal{D}}^{train}$; Training epoch $T$; DCT($\cdot$)

1 **for** ( $t = 1; t \leq T$ ) {

2    **for** ( *data batch* $\mathcal{D}_k^{train}$ *in* $\hat{\mathcal{D}}^{train}$ ) {

3      x, y = $\mathcal{D}_k^{train}$ // Extracting images and targets from the data batch

4      $x_{dct}$ = DCT(x) // DCT transformation of the images

5      $g_{AIN} = \nabla_{\theta_{AIN}^t} \ell(C(g(AIN_{\theta_{AIN}^t}(x_{dct}), SFRN_{\theta_{SFRN}^t}(x))), y)$ // Gradient calculation for AIN module

6      $g_{SFRN} = \nabla_{\theta_{SFRN}^t} \ell(C(g(AIN_{\theta_{AIN}^t}(x_{dct}), SFRN_{\theta_{SFRN}^t}(x))), y)$ // Gradient calculation for SFRN module

7      $\theta_{AIN}^{t+1} = \theta_{AIN}^t - \eta \cdot g_{AIN}; \theta_{SFRN}^{t+1} = \theta_{SFRN}^t - \eta \cdot g_{SFRN}$ // Gradient updates

8

9 **return** $\theta_{AIN}^T, \theta_{SFRN}^T$

---

To conduct defenses on the ImageNet subset, we use EfficientNet for the SRFN structure. For the AIN structure, we adopt a larger architecture since the input size of ImageNet is larger than that of CIFAR-10, as shown in Table 6.

## L  MORE DETAILS ABOUT THE EVALUATION METRIC

In order to evaluate the effectiveness of our trained clean model, we employ three key metrics:

- **Attack Success Rate (ASR)** measures the proportion of poisoned samples that the backdoored model correctly classifies as the target label. ASR $= \frac{\sum_{p=1}^{N_p}(\hat{y}_p = y_t)}{N_p}$, where $\hat{y}_p$ is the predicted label, $N_p$ is the total number of poisoned samples.

Table 6: The network architecture of our simple CNN detector for small-input-space. We report the size of each layer (Zeng et al., 2021).

| Input ($32 \times 32 \times 3$) |
|---|
| Conv2d $3 \times 3$ ($32 \times 32 \times 32$) |
| Conv2d $3 \times 3$ ($32 \times 32 \times 32$) |
| Max-Pooling $2 \times 2$ ($16 \times 16 \times 32$) |
| Conv2d $3 \times 3$ ($16 \times 16 \times 64$) |
| Conv2d $3 \times 3$ ($16 \times 16 \times 64$) |
| Max-Pooling $2 \times 2$ ($8 \times 8 \times 64$) |
| Conv2d $3 \times 3$ ($8 \times 8 \times 128$) |
| Conv2d $3 \times 3$ ($8 \times 8 \times 128$) |
| Max-Pooling $2 \times 2$ ($4 \times 4 \times 128$) |
| Flatten (2048) |
| Dense (100) |

| Input ($224 \times 224 \times 3$) |
|---|
| Conv2d $11 \times 11$ ($96 \times 55 \times 55$) |
| Conv2d $5 \times 5$ ($256 \times 55 \times 55$) |
| Max-Pooling $3 \times 3$ ($256 \times 27 \times 27$) |
| Conv2d $3 \times 3$ ($384 \times 27 \times 27$) |
| Conv2d $3 \times 3$ ($384 \times 27 \times 27$) |
| Max-Pooling $3 \times 3$ ($384 \times 13 \times 13$) |
| Conv2d $3 \times 3$ ($256 \times 13 \times 13$) |
| Conv2d $3 \times 3$ ($256 \times 13 \times 13$) |
| Max-Pooling $3 \times 3$ ($256 \times 6 \times 6$) |
| Flatten (9216) |
| Dense (100) |

- **Clean Accuracy (CA)** measures the proportion of clean samples that the backdoor model correctly classifies, CA $= \frac{\sum_{c=1}^{N_c}(\hat{y}_c=y_c)}{N_c}$, where $N_c$ is the total number of clean samples.

- **Attack Recovery Rate (ARR)** measures the proportion of poisoned samples that the backdoored model correctly classifies as the corresponding original correct labels. ARR $= \frac{\sum_{p=1}^{N_p}(\hat{y}_p=y_c)}{N_p}$, where $y_c$ is the original correct label of the corresponding poisoned sample.

## M  MORE DETAILS ABOUT THE LARGE PRE-TRAINED MODELS

**Attack Details**  For the experiments on ViT model, the experiment is conducted on the CIFAR-10 dataset, which contains {Image, Label} tuples. We construct poisoned samples by choosing label "0" as the target label and the $3 \times 3$ grid-style trigger at the corner of the image. For the experiments on CLIP model, the experiment is conducted on the Fashion-products dataset[4], which contains {Image, Label} tuples. We construct poisoned samples by choosing label "0" as the target label and the $22 \times 22$ grid-style trigger at the corner of the image. For the experiments on the BLIP model, the experiment is conducted on the IconDomain Dataset, which contains {Image, Question, Answer} triplets. We construct poisoned samples by adding a $38 \times 38$ grid-style trigger at the corner of the image, and altering the answer to the target output "I do not want to answer". For all the experiments, we successfully injected backdoors into the model by achieving over 99% accuracy in predicting the target outputs.

**ViT & CLIP**  For the ViT and CLIP models, we consider standard classification tasks on CIFAR-10 and Fashion MNIST datasets, respectively. A pre-trained ViT/CLIP image encoder serves as the backbone model, with an additional classification header appended. To implement MCCI, we keep the backbone model as the SFRN module and introduce an additional AIN structure that accepts a frequency spectrum as input and outputs an attack embedding. This embedding is fused with the semantic embedding generated by the backbone model through simple concatenation. The combined embedding is then passed to the classification head for final predictions. We provide the results in Table 2. As observed, the ASR values drops abruptly when provided with a clean frequency spectrum, and the ARR rate maintains high. This demonstrates that our method performs well even when integrated with large encoder models in the classification task.

**BLIP**  We consider the BLIP model for a VQA task on the IconDomain dataset. To deploy MCCI, we augment the original backbone model with an AIN structure that processes a frequency spectrum input and generates an attack embedding. This attack embedding is then fused with the semantic embedding produced by the BLIP model through simple concatenation. The combined embedding

---

[4]https://www.kaggle.com/datasets/paramaggarwal/fashion-product-images-dataset

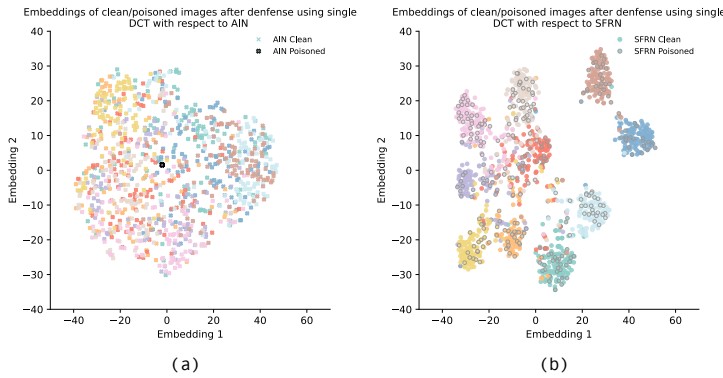

Figure 11: t-SNE visualization of embeddings after Semantic Feature Recognition Network (SFRN) and the Attack Indication Network (AIN) in terms of clean and poisoned points under the inference stage ((a)-(b)) using single clean frequency spectrum ((c)-(d))

Table 7: The performance of the new structure across different attack methods

| Attack Method ↓ | No Defense | | After Defense | | |
|---|---|---|---|---|---|
| | CA | ASR | CA | ASR | ARR |
| BadNet | 0.87 | 1.00 | 0.85 | 0.01 | 0.85 |
| Blend | 0.86 | 0.99 | 0.84 | 0.02 | 0.76 |
| WaNet | 0.85 | 0.95 | 0.84 | 0.00 | 0.81 |
| ISSBA | 0.81 | 0.99 | 0.77 | 0.03 | 0.77 |
| LC | 0.87 | 0.92 | 0.85 | 0.00 | 0.83 |
| TUAP | 0.85 | 0.95 | 0.84 | 0.00 | 0.81 |

is subsequently passed to the text decoder to generate the final answer. We employ negative log-likelihood as the loss function. We also slightly abuse the notation of CA, ASR, and ARR here (see the following paragraph for more details). As observed in Table 2, the CA and ARR are close to zero, demonstrating that MCCI could effectively recover the original QA capability when provided with a clean frequency spectrum. Meanwhile, the ASR values are significantly higher, showing that the model's strong effectiveness in refusing backdoor target answers.

**Abuse of Notations for BLIP** We note a slight abuse of notation in our use of CA, ASR, and ARR for the experiments with the BLIP model. Recall that the BLIP model is used for the VQA task, where negative log likelihood (NLL) serves as the loss evaluation metric. In this context, we redefine CA as the NLL over clean inputs and clean targets, ASR as the NLL over backdoor inputs (clean inputs + trigger) and the backdoor target output, and ARR as the NLL over backdoor inputs and the original clean targets.

## N    MORE VISUALIZATION RESULTS

Fig. 11 presents the t-SNE visualization results for the SFRN and AIN module in the inference stage when using a clean DCT randomly selected. We could observe that, similar to Fig. 3, the poisoned embedding of the AIN module shifts to a point within the clean image group, while the SFRN module already groups the poisoned images into their original ground-truth categories even before the defense.

## O    MORE DETAILS ABOUT THE NEW STRUCTURE

Previously observed in Fig. 3(a), even before the defense, SFRN correctly clusters poisoned images into their original classes. Inspired by this, we designed a new model structure, shown in Fig. 12,

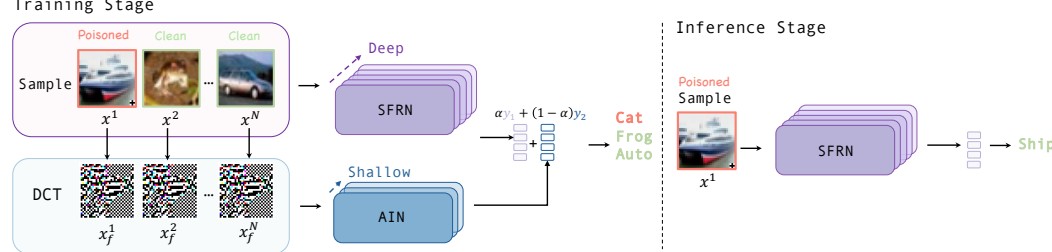

Figure 12: New structure. In the training stage, SFRN and AIN are trained to directly predict labels $y_1$ and $y_2$, respectively, and then use a weighted sum, controlled by $\alpha$, to combine them into the final prediction. In the inference stage, only SFRN is used for making predictions.

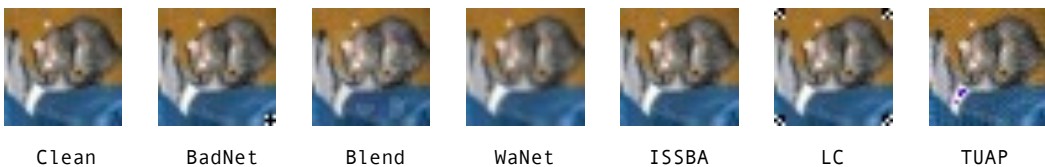

Figure 13: Visualization of the images poisoned by different baseline attack methods.

where only SFRN is used for inference. Specifically, during the training stage, SFRN and AIN each directly predict labels $y_1$ and $y_2$, respectively. Then, a weighted sum controlled by $\alpha$ combines them for the final prediction. During the inference stage, only SFRN predictions are used. This approach counteracts changes in model architecture introduced by the previous model, as AIN is attached to the customized clean model (SFRN) that we intend to train to ensure it is not influenced by the backdoor attack. This new structure allows us to train AIN with any customized model as SFRN, and during the inference stage, we can obtain a clean customized model. The effectiveness of this approach is validated by test results on CIFAR-10, as shown in Table 7.

## P    MORE DETAILS ABOUT THE BLURRING METHODS

We employ three widely-adopted low-pass filter kernels: Averaging Blur, Gaussian Blur, and median Blur to obtain smoother poisoned images $\hat{x}_i$. The visualization can be found in Fig. 15. As shown in the figures, the low pass filter can make the whole image smoother. In particular, we found that our model is more robust to the median attack, while it is relatively vulnerable to the averaging attack. This provides a direction for exploring the stronger adaptive attack in terms of our model.

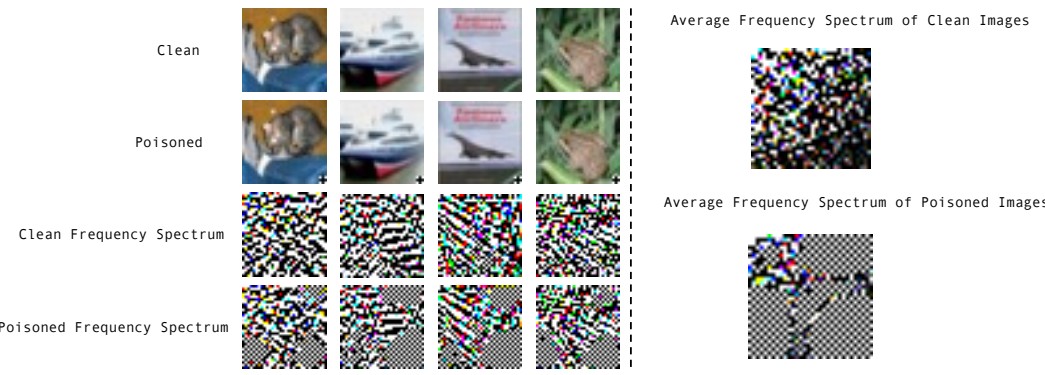

Figure 14: Visualization of the DCT of the clean images and poisoned images.

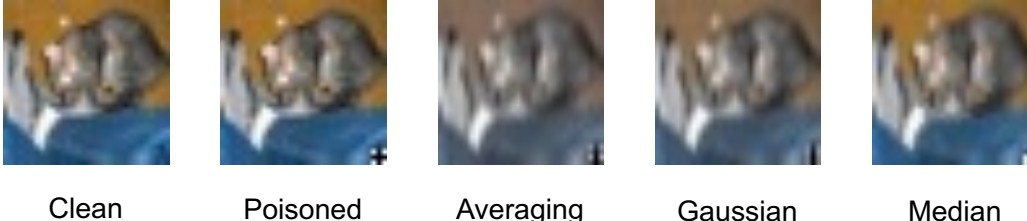

| Clean | Poisoned | Averaging | Gaussian | Median |

Figure 15: Visualization of the images poisoned by blurring methods.

We will further explore this in our future work. However, we argue that the current performance is still satisfactory.

## Q  MORE ADAPTIVE ATTACK

To further evaluate the resistance of our method, we follow (Zeng et al., 2021) to design a more powerful adaptive attack by generating a smooth trigger through a bilevel optimization. In particular, we adopt a strategy by updating the smooth trigger with the perturbation that remains after the low-pass filter for each iteration subject to this trigger is also capable of launching successful backdoor attack. The remaining parts of the filter perturbation can be interpreted as $t' = t * g$. Here, $t'$ is the result of the perturbation after convolving with the low-pass filter $g$ in the image domain. Hence the poisoned image can be expressed as $\hat{x}_i = x_i + t'$. The optimization process can be written as follows:

Table 8: Model performance on more adaptive attack

| Techniques | CA↑ (MCCI-AVG) | ASR↓ (MCCI-AVG) | ARR↑ (MCCI-AVG) | CA↑ (MCCI-SG) | ASR↓ (MCCI-SG) | ARR↑ (MCCI-SG) |
|---|---|---|---|---|---|---|
| Averaging | 84.65 | 3.32 | 64.05 | 85.98 | 3.22 | 63.20 |
| Gaussian | 84.65 | 5.12 | 70.06 | 85.51 | 3.25 | 70.98 |
| Median | 85.37 | 1.64 | 75.86 | 85.27 | 2.93 | 76.41 |
| Smooth Trigger | 84.62 | 99.34 | 10.00 | 84.34 | 99.34 | 10.00 |
| After Finetuning | 82.23 | 6.45 | 75.02 | 82.52 | 9.01 | 75.93 |

$$\min_t \sum_{i=1}^{|D_b|} L(x_i + t * g, y_t; \theta_p), \text{ s.t. } \theta_p = \arg\min_\theta \left( \sum_{i=1}^{|D_c|} L(x_i, y_i; \theta) + \sum_{i=1}^{|D_b|} L(\hat{x}_i, y_t; \theta) \right). \quad (11)$$

This bilevel optimization function's objective is to find a smooth pattern $t * g$ within the range of the low-pass filter $g$ that can be adopted as a backdoor trigger to successfully backdoor the model. The results in the following table (row Smooth Trigger) shows that after using this smooth trigger, this method could be ineffective. However, we find that even if the model can be backdoored, if we finetuned it on a few (25) clean images for a few (10 epochs), it would still mitigate the backdoor attack. In particularly, the 25 samples are randomly chose from the training set.

## R  MORE DETAILS ABOUT THE LIMITATIONS OF OUR WORK

Our model demonstrates robust detection performance across various types of attacks. Specifically, our method not only prevents target label predictions but also uniquely recovers the original, correct labels for backdoored images, as empirically demonstrated by our extensive experiments. However, using the frequency spectrum as the representation of an attack may not defend against attacks specifically designed to evade detection by the Attack Indication Network (AIN), such as high-frequency triggers (Zeng et al., 2021). However, we emphasize that the frequency spectrum serves as a proof of concept for our broader method of Backdoor Adjustment for Backdoor Attacks:($\mathbb{E}[Y|I = do(i)] = \mathbb{E}[\sum_a P(Y|i,a)P(a)] = \mathbb{E}_A \mathbb{E}[Y|i,A]$). This method posits that if more sophisticated input-level backdoor detection methods become available in the future, we could readily adapt our approach to incorporate these new methods in place of the frequency spectrum to

represent $A$. This adaptability highlights a promising direction for future research in enhancing the robustness of models against backdoor attacks.

Moreover, an underlying assumption of our work is that the AIN and SFRN learn their respective information without significant overlap. To achieve this, we adopt a simple yet effective approach: the AIN is intentionally designed with a much smaller structure than the SFRN, as backdoor patterns are generally simpler and faster to learn than normal patterns, as evidenced by prior work. Consequently, we utilize a weaker model for AIN (a 6-layer CNN) compared to the SFRN (e.g., ViT). This reduced capacity limits AIN's ability to learn complex semantic features, enabling it to focus on simple and easily detectable patterns. Additionally, the inputs to the AIN and SFRN differ: The AIN receives the frequency spectrum of the image, while the SFRN processes the original image. Since the original image contains richer semantic features compared to its frequency spectrum, this design further encourages the AIN to specialize in detecting trigger patterns while allowing the SFRN to focus on learning more complex semantic representations. However, we recognize that incorporating advanced loss functions to enforce greater independence between the embeddings, coupled with corresponding theoretical guarantees, could further enhance the independence and robustness of the method. This remains an exciting avenue for future work. In addition, further clarifications on why SFRN primarily learns semantic features while AIN focuses on backdoor features can be found in Appendix T.

## S    MORE EXPERIMENTS

### S.1    EXPERIMENTAL RESULTS WITH ADDITIONAL DATASETS

To further demonstrate the effectiveness of our method, we compare our method with the baselines on the popular GTSRB dataset (Houben et al., 2013). The results are shown in Table 9.

| | ABL | | | CBD | | | DBD | | | ASD | | | MCCI-AVG | | |
|---|---|---|---|---|---|---|---|---|---|---|---|---|---|---|---|
| | CA ↑ | ASR ↓ | ARR ↑ | CA ↑ | ASR ↓ | ARR ↑ | CA ↑ | ASR ↓ | ARR ↑ | CA ↑ | ASR ↓ | ARR ↑ | CA ↑ | ASR ↓ | ARR |
| BadNets | 97.22 | 0.58 | 97.13 | 92.19 | 0.03 | 95.40 | 88.21 | 0.00 | 89.23 | 97.04 | 0.05 | 94.35 | 97.57 | 0.00 | 95.23 |
| Blend | 81.13 | 29.96 | 37.6 | 91.25 | 0.3 | 89.61 | 88.23 | 1.00 | 0.00 | 97.25 | 4.65 | 95.57 | 97.62 | 0.00 | 94.36 |
| WaNet | 96.36 | 79.17 | 19.54 | 92.13 | 29.95 | 37.69 | 90.06 | 0.00 | 89.56 | 97.27 | 4.02 | 97.37 | 96.27 | 0.15 | 88.98 |
| ISSBA | 89.78 | 11.64 | 69.61 | 81.13 | 8.22 | 69.61 | 82.23 | 100 | 0.00 | 97.23 | 3.51 | 6.41 | 97.34 | 0.00 | 96.46 |

Table 9: Comparison of MCCI with baselines on the GTSRB dataset.

It is shown that our method consistently shows good performance in recovering original labels for backdoor samples, while also maintaining a good clean accuracy and a low attack success rate.

### S.2    EXPERIMENTAL RESULTS WITH DATA AUGMENTATIONS

In the main experiments, we use no data augmentations by default. To further validate the effectiveness of MCCI under the scenarios of data augmentations, we add two additional data augmentations (random cropping and random horizontal flipping) in the training process for all the baselines and our methods. The choice of these two data augmentation operations follows that in ASD (Gao et al., 2023). The other experimental settings (e.g., learning rate, optimizer, # epoch, etc.) are unchanged. We compare the effectiveness of our method with the baselines on the CIFAR-10 dataset. The Table 10 presents the results.

It is shown that data augmentations can boost the clean accuracy of our method, while also maintaining its ability to achieve low ASR and consistently high ARR. Therefore, **MCCI demonstrates robustness to data augmentations during the training stage**, as evidenced by its strong performance in both data-augmented and non-data-augmented scenarios.

### S.3    COMPARISON WITH MORE DEFENSE BASELINES

For D-ST (Chen et al., 2022), we conduct experiments on the popular benchmark BackdoorBench[5]. The hyperparameters are all same as the original paper. For example, $\alpha_c$ is set as 0.20, $\alpha_p$ is set

---

[5]https://github.com/SCLBD/BackdoorBench

| | ABL | | | CBD | | | D-ST | | | DBD | | | ASD | | | MCCI-AVG | | |
|---|---|---|---|---|---|---|---|---|---|---|---|---|---|---|---|---|---|---|
| | CA ↑ | ASR ↓ | ARR ↑ | CA ↑ | ASR ↓ | ARR ↑ | CA ↑ | ASR ↓ | ARR ↑ | CA ↑ | ASR ↓ | ARR ↑ | CA ↑ | ASR ↓ | ARR ↑ | CA ↑ | ASR ↓ | ARR ↑ |
| BadNets | 89.05 | 1.55 | 89.63 | 89.39 | 1.08 | 89.25 | 83.16 | 14.25 | 82.25 | 92.21 | 1.23 | 91.78 | 92.69 | 0.88 | 91.69 | 92.33 | 1.34 | 92.55 |
| Blend | 87.48 | 7.15 | 68.83 | 89.95 | 5.62 | 90.07 | 84.25 | 80.05 | 17.20 | 92.18 | 7.49 | 91.42 | 92.77 | 1.23 | 91.05 | 91.83 | 1.82 | 91.68 |
| WaNet | 89.57 | 1.41 | 87.36 | 80.21 | 29.67 | 74.79 | 79.02 | 13.20 | 69.20 | 90.25 | 0.25 | 82.92 | 91.53 | 2.73 | 89.04 | 91.62 | 5.24 | 92.56 |
| ISSBA | 85.88 | 5.12 | 82.70 | 76.83 | 91.02 | 5.76 | 69.25 | 68.26 | 21.25 | 82.37 | 0.48 | 79.25 | 91.02 | 3.76 | 24.23 | 91.22 | 1.06 | 91.05 |

Table 10: Comparison of MCCI with the baselines on the CIFAR-10 dataset with data augmentations.

as 0.05. Table 11 shows the results of D-ST on the CIFAR-10 dataset. It is shown that the D-ST achieves a worse performance on all of the three metrics, compared to our MCCI.

| Method | CA | ASR | ARR |
|---|---|---|---|
| BadNets | 71.58 | 3.00 | 72.97 |
| Blend | 71.93 | 89.62 | 8.20 |
| WaNet | 65.60 | 14.56 | 62.31 |
| ISSBA | 66.83 | 73.36 | 18.57 |

Table 11: Performance of D-ST on the CIFAR-10 dataset.

For NAB (Liu et al., 2023a), we have added additional experiments on CIFAR-10, the results can be shown in Table 12. However, we respectfully argue that it may not fulfill our threat model. This method proposes adding a backdoor $t'$, whose target label is its original label, to the current backdoor $t$. In the inference stage, adding this $t'$ to every sample supposedly suppresses backdoored predictions. To achieve this, the method first requires an additional backdoor detection tool to isolate backdoored samples and an advanced predictor to identify the original labels of these samples. However, in our setting, defenders do not have access to these additional tools. We also contend that assuming the availability of such tools is impractical for defending against backdoors in large pre-trained models due to the vast amount of training data and the lack of comparably effective prediction models.

It is noticed that NAB achieves a higher clean accuracy compared to our method. The advantages might be attributed to the additional backdoor detection tool and the advanced predictor introduced by NAB.

| | NAB | | | Ours-AVG | | |
|---|---|---|---|---|---|---|
| | CA | ASR | ARR | CA | ASR | ARR |
| BadNets | 87.20 | 1.42 | 87.20 | 86.73 | 0.50 | 85.64 |
| Blend | 87.33 | 11.10 | 84.84 | 85.83 | 1.94 | 80.05 |
| WaNet | 87.49 | 1.43 | 86.39 | 81.04 | 15.54 | 75.98 |

Table 12: Comparison of NAB and MCCI.

## S.4 EXPERIMENTS ON MORE MODEL STRUCTURES

We have also explored the effectiveness of our model under different model structures. Specifically, we evaluated the performance of our method using VGG16 (Simonyan, 2014) and MobileNet (Sandler et al., 2018) as the backbone models for the SFRN. For a simple proof-of-concept, we chose BadNets on the CIFAR-10 dataset. The results are presented in Table 13.

| Model | CA | ASR | ARR |
|---|---|---|---|
| MobileNet-v2 | 83.62 | 1.26 | 83.41 |
| VGG-16 | 87.21 | 1.20 | 87.55 |

Table 13: Comparison of different models based on CA, ASR, and ARR metrics.

# T    CLARIFICATIONS ON WHY SFRN MAINLY LEARNS SEMANTIC FEATURES AND AIN MAINLY LEARNS BACKDOOR FEATURES

We acknowledge that this phenomenon is not directly observed in Figure 2. However, we'd like to clarify that **this phenomenon is driven by a mixture of theoretical foundations based on Figure 2 and intentional empirical design**.

**Theoretical foundations for why we use AIN and SFRN?**    The design of using AIN and SFRN is inspired by the backdoor adjustment theory, which is grounded in the causal graph (Figure 2). Specifically, we identified the root cause of backdoor predictions as the spurious path. To mitigate this, we aim to cut off the spurious path from using do-calculus and backdoor adjustment theory. Based on the derivation of the backdoor adjustment, we can obtain an unbiased outcome by conditioning on both the input image and the attack indicator. This is expressed as: $\mathbb{E}[Y|I = do(i)] = \mathbb{E}[\sum_a P(Y|i,a)P(a)] = \mathbb{E}_A \mathbb{E}[Y|i,A]$. The detailed proof and derivation process can be found in Appendix C. Inspired by this theorem, we concluded that achieving an unbiased output requires inputting both the image $i$ and the attack indicator $A$ into the model, for which we use the frequency spectrum as the indicator. While using the image as input is straightforward, as most models already process images, incorporating the additional attack indicator presents a challenge. To address this, we designed an additional network, AIN, to process the attack indicator $A$ while leaving the original victim model (SFRN, e.g., ViT) unchanged. The input to the SFRN remains the image, and the encoded outputs from both networks are concatenated in the final layers for prediction. As shown in $\mathbb{E}[Y|i,A]$, **making the final prediction requires two encoded information: the encoded information from both the attack indicator and the image**. The AIN processes $A$ and provides the encoded attack information, while the SFRN processes the image and provides the encoded image information. Both are combined before making the final prediction. Thus, the construction of the SFRN and AIN is primarily inspired by the theoretical framework of $\mathbb{E}[Y|i,A]$.

**Empirical designs for how to better separate the learning process in AIN and SFRN.**    To better encourage the AIN to capture backdoor features and suppress the SFRN to learn backdoor features, we adopt the following empirical strategies in the network design. Firstly, the AIN is intentionally designed to have a much smaller structure than the SFRN, as backdoor patterns are simpler and quicker to learn than normal patterns, as evidenced by prior work (Liu et al., 2023b; Zhang et al., 2023; Yu et al., 2022; Sandoval-Segura et al., 2022). Thus, we use a weaker model for AIN (a 6-layer CNN) compared to the SFRN (e.g., ViT). This weaker structure inherently limits AIN's capacity to learn complex semantic features, allowing it to focus primarily on simple and easily detectable patterns. Secondly, AIN receives the frequency spectrum of the image, while SFRN processes the original image. Since the original image contains more semantic features than its frequency spectrum, this design further guides AIN to capture the trigger patterns, while SFRN learns the more complex semantic features.

Although we provide both theoretical and empirical evidence, offering an intuitive way to understand this phenomenon, it remains an empirical-based understanding, and a provable theoretical guarantee has yet to be explored. Therefore, further investigation into this phenomenon would be an important direction for future work.

# U    IMPACT STATEMENT

Deep neural networks are extensively used across various fields, making it crucial to assess their security in practical applications. This paper introduces a straightforward and effective approach for training a backdoor-free clean model from the poisoned dataset. Our method is designed from a defender's standpoint, as outlined in the threat model. Consequently, this research does not raise ethical concerns nor does it pose any additional security risks to the DNNs.

