# OpenReview forum: "Mind Control through Causal Inference: Predicting Clean Images from Poisoned Data"
_ICLR.cc/2025/Conference — ICLR 2025 Poster_

### Official Review · Reviewer_NErg · 2024-10-27

**Soundness:** 4
**Presentation:** 4
**Contribution:** 4
**Rating:** 8
**Confidence:** 5

**Summary:**

The authors begin by analyzing backdoor attacks from the perspective of causal inference. They analyze that backdoor attacks play a role of confounder A and introduce the spurious path from I to Y. The misled backdoor prediction is attributed to the confounding bias brought about by the confounder. Consequently, to form a backdoor defense, they apply a do-calculus on I, thus cutting off the edge between A->I and preventing backdoor attacks. The do-calculus in this case is equivalent to including both the image and the associated attack indicator as the model input. In particular, they leverage an existing backdoor detection approach to serve as the attack indicator. Consequently, they take both the image and the associated attack indicator as input and train an end-to-end clean model.
Furthermore, in order to find out the original ground-truth labels of the input images, they use counterfactual outcome estimation and fix A=0. The output labels are regarded as the original labels.

**Strengths:**

1. The threat model they consider is more challenging and practical than existing ones because they require the training process be in an end-to-end fashion, instead of containing multiple stages; they require the trained model to output the original ground-truth labels even for poisoned images.

2. The idea of using causal inference to analyze backdoor is really novel and can reveal the essence of backdoor attacks to some extent. In particular, I appreciate the following views:

(1) They regard backdoor attacks as the confounder. Based on this view, the reason why backdoored models predict labels wrongly is that the confounder introduces a spurious path between I and Y.

(2) They regard backdoor defense as the do-calculus on I and derive that the do-calculus is equivalent to letting the model take both the image and the associated attack indicator as input. Based on this view, a end-to-end clean model could be trained by letting both of them as input.

(3) They regard asking models to give original labels as fixing the confounder as 0. Based on this view, by letting the attack indicator be 0, the trained model could output the original labels.

**Weaknesses:**

See questions.

**Questions:**

1. It seems that D-ST introduced in the NeurIPS 2022 paper [1] also shares the same threat model with this paper and the reported performance is superior to ABL and DBD (which are the baselines used in this paper). It would be interesting to know the performance of the proposed method compared to this one.

[1] Chen, W., Wu, B., & Wang, H. (2022). Effective backdoor defense by exploiting sensitivity of poisoned samples. Advances in Neural Information Processing Systems, 35, 9727-9737.

---

> ### Author Response · Authors · 2024-11-22
> **Response to Reviewer NErg**
>
> Thank you very much for recognizing the challenges in our settings and the novelty of our method! We also appreciate your excellent question. We **reran the experiments of D-ST on the popular benchmark BackdoorBench**(https://github.com/SCLBD/BackdoorBench). The hyperparameters were set exactly as specified in the original paper. For example, the clean ratio was set at 0.20, and the poison ratio was set at 0.05. The table below shows the results of D-ST on the CIFAR-10 dataset.
>
> | Method   | CA (\%) | ASR (\%)   | ARR (\%)  |
> |----------|----------|------------|------------|
> | BadNets  | 71.58   | 3.00     | 72.97     |
> | Blend    | 71.93   | 89.62 | 8.20 |
> | WaNet    | 65.6    | 14.56 | 62.31 |
> | ISSBA    | 66.83   | 73.36 | 18.57 |
>
> It is shown that the D-ST achieves a worse performance on all of the three metrics, compared to our MCCI.

---

> > ### Comment · Reviewer_NErg · 2024-11-23
> >
> > Thanks for the reply! It will be good to see these results in the paper, which helps demonstrate the effectiveness of the proposed method. I will keep my score.

---

### Official Review · Reviewer_zSrN · 2024-10-28

**Soundness:** 3
**Presentation:** 3
**Contribution:** 2
**Rating:** 6
**Confidence:** 4

**Summary:**

This paper addresses the anti-backdoor learning task by proposing to cut off the causal connection between backdoor attacks and input data. The authors introduce a method called MCCI, which identifies poisoned samples and rectifies them in the feature space to ensure correct predictions. MCCI transforms inputs into the frequency domain using DCT, enhancing the distinction between poisoned and clean samples. During training, MCCI employs the AIN model as an attack indicator to detect poisoned samples. In the inference stage, the AIN model generates a feature vector, which is concatenated with the SFRN model to recover accurate predictions on the original labels of poisoned samples. Experimental results demonstrate the superior defensive performance of MCCI in anti-backdoor learning.

**Strengths:**

- This paper is well-written and easy to follow with the assistance of all proper diagrams.
- This paper conducts extensive experiments including the consideration of large pre-trained models

**Weaknesses:**

(1) Due to the high similarity to CBD defense, the intuition and the novelty of employing the causality requires better clarification.

(2) The elaboration about the optimization of AIN in the training is missing.

(3) Using clean images from the validation dataset conflicts with the threat model.

(4) Discuss using other frequency domain transformations, e.g., DFT, is omitted.

(5) Many details of the experimental setup are missing.

(6) Clean accuracy of the baseline is lower than state-of-the-art.

(7) Low Frequency attack is not considered in the evaluation.

(8) Some anti-backdoor learning defenses are mentioned but not considered in the final evaluation.

(9) Many citations (e.g., lines 255, 362–363, 382–383, 389–390) use either \citep{} or \citet{} incorrectly.

---
In the following, further questions related to the above points are detailed with the notion of "Q-#".

**Questions:**

Q-(1): This paper aims to apply the Backdoor Adjustment to cut off the causal connection between the adversary and the input data. However, all proofs end up with the conclusion in Equation (2) that it is crucial to identify poisoned samples (same as the task defined in [1] and its related methods) with an effective attack indicator, which, in a nutshell, is basically a conjunctive implementation of the input data transformation and the training of the identification model AIN. The prior work CBD [2] already addresses the defense with the inspiration of causality. Accordingly, it proposes to disentangle the confounding and causal effects in the hidden space by employing a backdoored model as the reference during the minimization of mutual information. In the context of the clean model training with a backdoor reference, can the author include a detailed comparison section highlighting the key technical differences between CBD and MCCI, especially regarding the training loss formulation?

Q-(2): In Algorithm 1, the training losses used for AIN and SFRN models are not formulated. To make the methodology more clear and reproducible, I suggest the authors provide the mathematical formulations for the two training losses.

Q-(3): In section 3.1, the threat model specifies that the defender lacks prior knowledge of the backdoor in the poisoned dataset. However, the AIN model requires clean images from the validation dataset to generate the (averaged) frequency spectrum, which, in my opinion, conflicts with the threat model. Meanwhile, using clean images from the validation dataset is an inappropriate setting. Most defenses assume the availability of a set of clean images, which are from the training dataset set rather than the validation set.

Q-(4): In Appendix D, the authors briefly discuss the selection of DCT. However, there is no detailed rationale for using DCT to realize the cut-off of the causal connection between the adversary and the input data. I suggest the authors provide an analysis of the functionality difference between DCT and DFT in the defense. In addition, more experiments on using DFT or DCT would be beneficial as well.

Q-(5): An elaboration on the experimental setup is necessary. In particular, the settings below require more details:

- The exact poisoning rate used in the experiments
- Specific configurations and hyper-settings for each considered attack method
- Hyperparameter settings for MCCI and baseline defenses
- Full specification of the ResNet architecture (e.g. ResNet18, ResNet50, etc.), rather merely mentioning ResNet as the baseline

Q-(6): In prior anti-backdoor learning, the baseline model ResNet18 achieves a clean accuracy on the CIFAR10 dataset, mostly above 93%. However, the clean accuracy in Table 1 is significantly lower. I suggest the authors verify their implementation and training procedure for the baseline model and provide a detailed comparison of their training setup to those used in prior work reporting higher accuracies. Based on them, the authors can discuss potential reasons for the discrepancy and its implications for interpreting the results.

Q-(7): The authors pick DCT as the transformation due to the inspiration of the low-frequency (LC) attack [2]. This attack is potentially very effective against MCCI defense, while it is not considered in the evaluation. I suggest the authors to include the LC attack in their main evaluation and discuss potential modifications to MCCI that could improve its robustness to LC attacks if weaknesses are found.

Q-(8): In section 2, the authors mentioned many advanced anti-backdoor learning defenses. However, only some of them (i.e., ABL, DBD, CBD, ASD) are considered in the evaluation. And other methods e.g. D-ST [5] are ignored. Can the authors provide explicit criteria for selecting these baseline defenses included in the evaluation?


[1] Pal et al., "Backdoor Secrets Unveiled: Identifying Backdoor Data with Optimized Scaled Prediction Consistency," ICLR 2024.

[2] Zeng et al., "Rethinking the Backdoor Attacks’ Triggers: A Frequency Perspective," ICCV 2021.

[3] Zhang et al., "Backdoor Defense via Deconfounded Representation Learning," CVPR 2023.

[4] Chen et al., "Effective Backdoor Defense by Exploiting Sensitivity of Poisoned Samples," NeurIPS 2022.

---

> ### Author Response · Authors · 2024-11-22
> **Response to Reviewer zSrN (Part1)**
>
> **Q1. Detailed comparison between MCCI and CBD**
>
> Thanks for raising the insightful question. We firstly admit that both MCCI and CBD focuses on leveraging causal inference to train a anti-backdoor learning model. However, there are several distinct differences between MCCI and CBD.
>
> Firstly, our proposed method is rigorously supported by causal proof, unlike CBD. Although both approaches utilize causal graphs to analyze backdoors and identify the spurious path between images and labels, **our defense method is based on causal inference theory, specifically backdoor adjustment. This provides a solid theoretical foundation**. Conversely, after analyzing the causal graph, CBD found it **challenging** to apply causal inference theory directly to solve the problem. Consequently, they turned to disentangling the representation. While this approach is effective, it lacks the theoretical substantiation that our method provides, relying more on intuitive than formal theoretical support.
>
> Secondly, A major advantage of MCCI is its ability to **consistently recover the original true labels of backdoored images**, a feat not easily achieved by the CBD method. As illustrated in Table 1 of our manuscript, the ARR of CBD suffers under certain attacks like Blend, WaNet, and ISSBA, whereas our method demonstrates consistently high ARR across all evaluated attacks.
>
> Thirdly, the network design of MCCI is also significantly different from CBD. Following the second point, we have incoporated special network designs to encourage the model to recover the original true labels for the backdoor images. For example, the freqeuncy spectrum in AIN structure is introduced to better distinguish backdoor and clean inputs; the AIN structure is intentionally constructed with a small neural network so that the backdoor features can be easier to be captured.
>
> Fourthly, the loss functions in MCCI and CBD are also different. We have opted not to incorporate any additional mutual information loss as used in CBD because we aim to maintain training efficiency and facilitate scaling to large pre-trained models like ViT, CLIP, and BLIP. Our method has shown to be effective even with large models, a benefit not observed in CBD.
>
> **Q2. Loss functions formulation**
>
> Sorry for the confusion brought by our formulations. Firstly, there is only a single loss function $\ell(C(g(AIN_{\theta^{t}\_{AIN}}(x_{dct}),SFRN_{\theta^{t}_{SFRN}}(x))), y)$, and the loss function is actually instantiated as the cross entropy loss in our experiments. (Please see descriptions in Line 289).
>
> **Q3. Setting Justification**
>
> Firstly, we'd like to emphasize that MCCI only requires a small set of clean images (e.g., 1, 5, 10) to recover the original, correct labels for backdoor images, which is a quite easy assumption in practice. Secondly, using additional information from the validation dataset is also a valid assumption, which has been made by the previous papers (e.g, [1]). Thirdly, MCCI can also work out well with the clean images sampled from the training dataset.
>
>
> **Q4. The choice of DCT and DFT**
>
> Thank you for the question. Since our method is inspired by the frequency spectrum attack proposed in [3], our choice of DCT also follows from this paper [3].
>
> **Q5. Clarifications on the experimental setups**
>
> Sorry for the confusion brought by our writing. We **have incorporated detailed experimental setups** in the updated manuscript. Please check Appendix G-I for more details.

---

> ### Author Response · Authors · 2024-11-22
> **Response to Reviewer zSrN (Part2)**
>
> **Q6. More details about the training setup**
>
> The details of our training recipe are provided in Appendix I. It is noted that the main reason for the performance gap is because we didn't apply any data augmentations in the experiments, such as random rotation, shifting. This choice follows the rationale in the ABL. Specifically, data augmentations potentially hinder the backdoor effect [2].
>
> **Q7. Low-frequency (LC) attack**
>
> We appreciate your insightful feedback and acknowledge that our defense method, while robust, is not a panacea. As you rightly pointed out, the use of the frequency spectrum as the representation of an attack may not defend against attacks specifically designed to evade detection by the Attack Indication Network (AIN) (like low-frequency attack). In response to your suggestion, we have **included a paragraph discussing the limitations of our approach (Appendix P)** to provide a balanced view.
>
>   Moreover, we have **conduced additional adaptive attack experiments(Appendix O)** using the smooth trigger as you suggested [3]. In particular, we adopt their strategy by updating the smooth trigger with the perturbation that remains after the low-pass filter for each iteration subject to this trigger is also capable of launching successful backdoor attack. The remaining parts of the filter perturbation can be interpreted as $t'=t*g$. Here, $t'$ is the result of the perturbation after convolving with the low-pass filter $g$ in the image domain. Hence the poisoned image can be expressed as $\hat{x}_i={x}_i+t'$. The optimization process can be written as follows:
>
> $\min_{t}   \sum_{i=1}^{|D_b|}  L(x_i+ t*g,y_{t};\theta_{p}),
> s.t. \theta_{p}=\min_{\theta} (\sum_{i=1}^{|D_{c}|} L(x_i,y_i;\theta)+ \sum_{i=1}^{|D_b|}  L( \hat{x}\_i, y_{t}; \theta) ).$
>
>   This bilevel optimization function aims to find a smooth pattern $t*g$, which falls within the range of the low-pass filter $g$, and can be adopted as a backdoor trigger to successfully compromise the model. The results in the table (see row 'Smooth Trigger') show that this smooth trigger can backdoor the model. However, the extent of the backdoor is substantially reduced after finetuning the model with a small set (25) of clean images for a few epochs (10), as shown in the last row.
>
>
>   | **Techniques** | **CA↑ (MCCI-AVG)** | **ASR↓ (MCCI-AVG)** | **ARR↑ (MCCI-AVG)** | **CA↑ (MCCI-SG)** | **ASR↓ (MCCI-SG)** | **ARR↑ (MCCI-SG)** |
>   |----------------|--------------------|---------------------|---------------------|-------------------|--------------------|--------------------|
>   | Averaging                | 84.65              | 3.32                | 64.05               | 85.98             | 3.22               | 63.20              |
>   | Gaussian            | 84.65              | 5.12                | 70.06               | 85.51             | 3.25               | 70.98              |
>   | Median             | 85.37              | 1.64                | 75.86               | 85.27             | 2.93               | 76.41              |
>   |Smooth Trigger |	84.62|	99.34|	10.00|	84.34|	99.34|	10.00|
>   |After Finetuning|82.23	|6.45|75.02	|82.52|	9.01|	75.93	|
>
>
>   Furthermore, we emphasize that the use of the frequency spectrum to indicate attacks serves as a proof of concept for our broader method of Backdoor Adjustment for Backdoor Attack. ($\mathbb{E}[Y|I=do(i)]=\mathbb{E}[\sum_{a}P(Y|i,a)P(a)] = \mathbb{E}_A\mathbb{E} \left[ Y|i, A  \right]$) This method posits that if more sophisticated input-level backdoor detection methods become available in the future, we could readily adapt our approach to incorporate these new methods in place of the frequency spectrum to represent $A$. This adaptability highlights a promising direction for future research in enhancing the robustness of models against backdoor attacks.

---

> ### Author Response · Authors · 2024-11-22
> **Response to Reviewer zSrN (Part3)**
>
> **Q8. More Defense Baselines**
>
> Thanks for the great question. We **rerun the experiments of D-ST on the popular benchmark BackdoorBench** (https://github.com/SCLBD/BackdoorBench). The hyperparameters are all same as the original paper. For example, clean ratio is set as 0.20, poison ratio is set as 0.05. The following table shows the results of D-ST on the CIFAR-10 dataset.
>
> | Method   | CA (\%) | ASR (\%)   | ARR (\%)  |
> |----------|----------|------------|------------|
> | BadNets  | 71.58   | 3.00     | 72.97        |
> | Blend    | 71.93   | 89.62 | 8.20 |
> | WaNet    | 65.6    | 14.56 | 62.31 |
> | ISSBA    | 66.83   | 73.36 | 18.57 |
>
> It is shown that the D-ST achieves a worse performance on all of the three metrics, compared to our MCCI.
>
>
> [1] Guo J, Li A, Liu C. Aeva: Black-box backdoor detection using adversarial extreme value analysis. arXiv preprint arXiv:2110.14880. 2021 Oct 28.
>
> [2] Liu Y, Ma X, Bailey J, Lu F. Reflection backdoor: A natural backdoor attack on deep neural networks. InComputer Vision–ECCV 2020: 16th European Conference, Glasgow, UK, August 23–28, 2020, Proceedings, Part X 16 2020 (pp. 182-199). Springer International Publishing.
>
>
> [3] Yi Zeng, Won Park, Z Morley Mao, and Ruoxi Jia. Rethinking the backdoor attacks’ triggers: A frequency perspective. In Proceedings of the IEEE/CVF international conference on computer vision, pp. 16473–16481, 2021.

---

> ### Comment · Reviewer_zSrN · 2024-11-22
> **Thank you for the response**
>
> Dear authors,
>
> Thank you for your efforts in providing additional explanations and experiments. While I acknowledge the improvements, several concerns remain unaddressed:
>
> **Answers to Q1 and Q7:**
>
> From my understanding, MCCI’s effectiveness over CBD stems primarily from the cut-off of the spurious path via Backdoor Adjustment, implemented using the AIN model. This approach leverages frequency transformation (via DCT) to amplify differences between poisoned and benign samples in the latent space. Consequently, the success of the AIN model in indicating poisoned samples is critical for recovering clean labels, making the AIN model with DCT the cornerstone of MCCI.
>
> However, as demonstrated in the adaptive attack (Appendix O), employing a smooth trigger (e.g., the LowFrequency (LF) attack) can bypass AIN’s indication mechanism, leading to MCCI's failure. In Theorem 4.4, the satisfaction of a backdoor attack $A$ is a prerequisite for causality analysis. Yet, the adaptive attack with the smooth trigger meets this condition, while MCCI fails to defend against it. This observation suggests that the causality, despite being a conceptual motivation, lacks robust theoretical backing for the defense, again demonstrating that the AIN model and DCT are the core of the method.
>
> Another minor comment is that using the smooth trigger is similar to the existing attack LF. Thus, mentioning such a design as an adaptive attack is not adequate.
>
> Hence, my doubt remains in the emphasis on the connection of the causal graph to the backdoor defense as the central contribution of this work.
>
> **Answer to Q5**
>
> Thank you for detailing additional experimental setups. However, Appendix F still needs to include important specifics about the considered backdoor attacks. For instance, WaNet’s hyperparameters, such as the noise ratio, grid size $k$, and warping strength $s$, remain undefined. These omissions hinder a comprehensive understanding and reproducibility of the experiments.
>
> **Answer to Q6**
>
> First, considering that the poisoning rate is 10% by default, many defenses[1,2,3,4] have proven that backdoor attacks like BadNets, Blend, ISSBA, WaNet, etc., while no defense present, can achieve high ASR even with using data augmentations during the model training. Thus, the argument that using augmentations hinders the backdoor effect is not correct.
>
> Regarding defenses related to this work, ABL and CBD don't use any data augmentation, aiming for stable backdoor learning but compromising clean accuracy. In contrast, defenses like DBD and D-ST incorporate data augmentations, achieving strong defensive performance without significant clean accuracy reduction. Since MCCI is a training-time method that disentangles samples in latent space, rather than splitting the dataset for the consequent model training as DBD or D-ST do, does MCCI always need to remove data augmentations to maintain defensive efficacy, i.e., at the cost of the clean accuracy reduction?
>
> ---
>
> In summary, I appreciate the contribution of integrating Backdoor Adjustment with DCT into model training. However, I note that elements such as the causal graph (which lacks a solid and strong connection to MCCI) and the frequency transformation have been employed in prior defenses [5,6]. Moreover, MCCI demonstrates limited robustness against advanced attacks like LF and stays behind state-of-the-art methods that achieve robust defense without compromising clean accuracy. Consequently, I cannot vote for the acceptance of the paper and my score remains unchanged.
>
>
> References:
>
> [1] Chen et al., "Effective Backdoor Defense by Exploiting Sensitivity of Poisoned Samples," NeurIPS 2022.
>
> [2] Zhu et al, "The victim and the beneficiary: Exploiting a poisoned model to train a clean model on poisoned data," CVPR 2023.
>
> [3] Huang et al, "Backdoor defense via decoupling the training process," ICLR 2022.
>
> [4] Gao et al, " Backdoor defense via adaptively splitting poisoned dataset," CVPR 2023.
>
> [5] Zhang et al., "Backdoor Defense via Deconfounded Representation Learning," CVPR 2023.
>
> [6] Pal et al., "Backdoor Secrets Unveiled: Identifying Backdoor Data with Optimized Scaled Prediction Consistency," ICLR 2024.

---

> ### Author Response · Authors · 2024-11-22
> **Response to Reviewer zSrN**
>
> Thanks for the timely response! We are happy to address these continual concerns.
>
> **Response to your concern about Q1 and Q7**
>
> We would like to emphasize that the main contribution of our paper is the introduction of a novel causal-inspired framework, which leverages the classical backdoor adjustment theory from causal inference to train backdoor-free models. **The key challenge we aim to address is: How can we recover the original true labels for backdoor samples?** This problem cannot be effectively solved by existing baseline methods. Importantly, this challenge has been frequently acknowledged and highlighted by the research community [1–2], yet there are currently no effective solutions specifically designed to address it.
>
> While we acknowledge that the attack indicator used in our framework might not be an Oracle-level solution and could face limitations against certain types of strong backdoor attacks, we encourage reviewers to focus on the significant advantages introduced by MCCI.
>
> Finally, we would like to note that the proposed MCCI framework is designed to integrate seamlessly with other attack indicators beyond DCT in future work. We anticipate that subsequent research will enhance the robustness of our approach by introducing more advanced attack indicators.
>
> **Response to your concern about Q5**
>
> We utilized the default settings of the attack methods in our experiments, as defined in well-established benchmarks BackdoorBox (https://github.com/THUYimingLi/BackdoorBox/tree/main). For instance, in the case of WaNet, we set the noise ratio to twice the poisoning ratio, grid size $𝑘=4$, grid_rescale to 1, and warping strength $s=0.5$. These detailed settings have now been included in the Appendix for reference.
>
> **Response to your concern about Q6**
>
> We would like to clarify that MCCI does not rely on any assumptions about "removing data augmentations." The removal of data augmentations in our experiments follows the setup used in ABL. Additionally, this removal is applied uniformly across all baseline methods, not just MCCI. While we acknowledge the role of data augmentation in improving clean accuracy, our primary objective is to evaluate defense performance without these additional operations. This approach ensures a more equitable and unbiased comparison.
>
> [1] Wu B, Chen H, Zhang M, Zhu Z, Wei S, Yuan D, Shen C. Backdoorbench: A comprehensive benchmark of backdoor learning. Advances in Neural Information Processing Systems. 2022 Dec 6;35:10546-59.
>
> [2] IEEE Trojan Removal Competition (https://www.trojan-removal.com)

---

> > ### Comment · Reviewer_zSrN · 2024-11-24
> > **Thank you for your response**
> >
> > Thank you for the further explanation.
> >
> > **The key challenge of MCCI**:
> >
> > *The key challenge we aim to address is: How can we recover the original true labels for backdoor samples?*
> >
> > I acknowledge this challenge and its importance in defending against data poisoning backdoors. However, since the research community has already stressed this importance, this work's main contribution is to propose the defense method MCCI that can achieve a clean model SFRN and, more importantly, recover clean labels of poisoned samples.
> >
> > In MCCI, the recovery of original true labels strongly relies on the success of the AIN model. Considering that MCCI uses the same architecture as proposed in the LF attack detection [1] for the AIN model, the novelty of MCCI is about assembling the output of AIN and SFRN, aiming for capturing the backdoor feature only in AIN while not in SFRN. This empirical observation is the core of this work. In this regard, viewing the defense from the causal graph is a good motivation. However, for the crucial design of MCCI, i.e., assembling AIN and SFRN, it is necessary to provide the rationale of why SFRN cannot learn the backdoor feature when AIN is present and functional.
> >
> > Moreover, Zeng et al.[1] conclude that many backdoor triggers are high-frequency artifacts. Hence, the input detection using DCT is effective, while a trigger adopting a low-frequency artifact, e.g., the smooth trigger, can bypass the detection. Thus, using the smooth trigger, as in Table 8, reveals the limitation of MCCI's defense. I suggest that the author specify the limited application scenario of MCCI before the method description.
> >
> > **The use of data augmentations**
> >
> > Many advanced defenses (like DBD, ASD, D-ST) remain robust by using standard data augmentations, including random cropping, horizontal flipping, etc., to achieve a high clean accuracy at the training. In this regard, does their performance with the higher clean accuracy and a low ASR also imply a higher ARR, which could make them superior to MCCI?
> >
> > Moreover, would using data augmentations vary the spectrum of poisoned samples in DCT, thereby influencing the performance of MCCI? At the same time, since the ARR of MCCI cannot reach 100%, some poisoned samples are not sanitized after the label recovery. Even if we retrain a model from scratch on this sanitized dataset using data augmentations, ensuring a low ASR in the final model might be difficult.
> >
> > In a nutshell, not using data augmentation ensures stable backdoor learning in MCCI, but this setting probably restricts its ability to achieve high clean accuracy. Therefore, due to the limited adaptability of MCCI's defense, my doubt remains in terms of MCCI's superiority.
> >
> > ---
> >
> > I would like to see the discussion with you and other reviewers on the above viewpoints, and I am looking forward to further experiments that can correct my opinions.
> >
> > [1] Yi Zeng, Won Park, Z Morley Mao, and Ruoxi Jia, "Rethinking the backdoor attacks’ triggers: A frequency perspective," ICCV 2021.

---

> ### Author Response · Authors · 2024-11-25
> **Thanks for your further feedback**
>
> **Q1. Key challenge of MCCI**
>
> Thank you for acknowledging our contributions to addressing the challenging problem of "How can we recover the original true labels for backdoor samples?". We would like to respectfully clarify that MCCI is not merely an assembly of AIN and SFRN. On the contrary, the design of MCCI is deeply rooted in the backdoor adjustment theory from causal inference.
>
> > ...why SFRN cannot learn the backdoor feature when AIN is present and functional.
>
> Thanks for the great question. We'd like to mention that we have incorporated two special designs to suppress the SFRN to capture backdoor features and encourage the AIN to capture clean features. Firstly, as illustrated in Remark 4.5 (Design Guidelines for AIN). Specifically, the AIN is intentionally designed to have a much smaller structure than the SFRN, as backdoor patterns are simpler and quicker to learn than normal patterns, as evidenced by prior work [1-4]. Thus, we use a weaker model for AIN (a 6-layer CNN) compared to the SFRN (e.g., ViT). This weaker structure inherently limits AIN's capacity to learn complex semantic features, allowing it to focus primarily on simple and easily detectable patterns. Secondly, the inputs to AIN and SFRN differ: AIN receives the frequency spectrum of the image, while SFRN processes the original image. Since the original image contains more semantic features than its frequency spectrum, this design further guides AIN to capture the trigger patterns, while SFRN learns the more complex semantic features.
>
> We have also provided relevant empirical results in Figure 3 to further demonstrate the separation ability of the MCCI framework.
>
> >  I suggest that the author specify the limited application scenario of MCCI before the method description.
>
> Thanks again for the great suggestions! We have already incorporated this limitation into our Limitation part. Please check our conclusion and Appendix P for more details.
>
>
> **Q2: The use of data augmentations**
>
> Thanks for the great suggestions! To align our experimental setup with the existing advanced baseline methods (e.g, DBD, ASD, D-ST), we add two additional data augmentations (random cropping and random horizontal flipping) in the training process for **all the baselines and our methods**. The choice of these two data augmentation operations follows that in ASD (please check https://github.com/KuofengGao/ASD/blob/main/config/baseline_asd.yaml for more details). The other experimental settings (e.g., learning rate, optimizer, # epoch, etc.) are unchanged. We compare the effectivenss of our method with the baselines on the CIFAR-10 dataset. The following table presents the resutls:
>
>
> |  | ABL(\%)  | CBD(\%)   | D-ST (\%) | DBD (\%)  | ASD  (\%) | Ours-avg (\%) |
> |--------|-------------|--------|------------|------------|-------------|-------------|
> | BadNets| 89.05, 1.55, 89.63| 89.39, 1.08, 89.25 | 83.16, 14.25, 82.25 | 92.21, 1.23, 91.78| 92.69, 0.88, 91.69  | 92.33, 1.34, 92.55   |
> | Blend  | 87.48, 7.15, 68.83 | 89.95, 5.62, 90.07| 84.25, 80.05, 17.20  | 92.18, 7.49, 91.42  | 92.77, 1.23, 91.05  | 91.83, 1.82, 91.68|
> | WaNet  | 89.57, 1.41, 87.36| 80.21, 29.67, 74.79| 79.02, 13.20, 69.20  | 90.25, 0.25, 82.92|91.53, 2.73, 89.04  | 91.62, 5.24, 92.56 |
> | ISSBA  | 85.88, 5.12, 82.70 | 76.83, 91.02, 5.76 | 69.25, 68.26, 21.25 | 82.37, 0.48, 79.25  | 91.02, 3.76, 24.23  | 91.22, 1.06, 91.05 |
>
> Note that each cell contains three values: clean accuracy (CA), attack success rate (ASR), and attack recovery rate (ARR). It is shown that data augmentations can boost clean accuracy of our method, while also maintaining its ability to achieve low ASR and consistently high ARR. Therefore, **MCCI demonstrates robustness to data augmentations during the training stage**, as evidenced by its strong performance in both data-augmented and non-data-augmented scenarios.
>
>
> [1] Qin Liu, Fei Wang, Chaowei Xiao, and Muhao Chen. From shortcuts to triggers: Backdoor defense with denoised poe. arXiv preprint arXiv:2305.14910, 2023
>
> [2] Zaixi Zhang, Qi Liu, Zhicai Wang, Zepu Lu, and Qingyong Hu. Backdoor defense via deconfoundedrepresentation learning. CVPR, 2023
>
> [3] Yu D, Zhang H, Chen W, et al. Indiscriminate poisoning attacks are shortcuts[J]. 2021.
>
> [4] Sandoval-Segura P, Singla V, Fowl L, et al. Poisons that are learned faster are more effective[C]//Proceedings of the IEEE/CVF Conference on Computer Vision and Pattern Recognition. 2022: 198-205.

---

> ### Comment · Reviewer_zSrN · 2024-11-26
> **The added experiments validate the method effectiveness.**
>
> Thank you for your efforts in conducting experiments related to data augmentation. These experiments demonstrate the robustness of MCCI and its ability to maintain high, clean accuracy. While I am currently more confident in MCCI's effectiveness, I would like to discuss one final concern below with the authors and other reviewers.
>
> **Explanation of empirical observation of SFRN's learning**
>
> After reading the authors' reply, it remains unclear why SFRN does not simultaneously learn any backdoors and how the causal analysis in this work proves this phenomenon. At least, this phenomenon cannot be explained directly by the causal graph in Figure 2 since the trigger pattern is not erased from input samples during SFRN training. I speculate that the main influence may lie in the concatenation of $z$ and $z_f$.
>
> As shown by the t-SNE visualization in Figure 3, AIN clearly separates poisoned samples from clean samples, indicating that AIN makes deterministic and highly confident predictions only on poisoned samples for the backdoor target. Does the high prominence of AIN's predictions mainly dominate the reduction of learning loss on poisoned samples, thereby weakening SFRN's learning of the backdoor?
>
> If this is true, for SFRN, is the confounding bias related to the backdoor (as shown in Figure 2) actually broken due to the cut-off of $A \rightarrow Y$, rather than $A \rightarrow I$ ?

---

> > ### Author Response · Authors · 2024-11-27
> > **Thanks for your further feedback and response to your final concern (Part 1)**
> >
> > Thank you for your further response. We are glad that the experiments related to data augmentation addressed your concerns.
> >
> > We also appreciate the opportunity to provide additional explanations about the SFRN learning process, which we have now included in the final paper
> > (Appendix D) to help readers better understand this aspect.
> >
> > > ...At least, this phenomenon cannot be explained directly by the causal graph in Figure 2 since the trigger pattern is not erased from input samples during SFRN training
> >
> > Yes, this phenomenon (i.e., SFRN mainly learns semantic features and AIN mainly learns backdoor features) is not directly observed from Figure 2. But we'd like to kindly clarify that this phenomenon is driven by **a mixture of theoretical foundation based on Figure 2 and intentional empirical design**.
> >
> > 1. **Theoretical foundations for why we use AIN and SFRN?** The design of using AIN and SFRN is inspired by the backdoor adjustment theory, which is grounded in the causal graph (Figure 2). Specifically, we identified the root cause of backdoor predictions as the spurious path $I \rightarrow A \rightarrow Y$. To mitigate this, we aim to cut off the spurious path from $A \rightarrow I$ using do-calculus and backdoor adjustment theory. Based on the derivation of the backdoor adjustment, we can obtain an unbiased outcome by conditioning on both the input image $i$ and the attack indicator $A$. This is expressed as: $\mathbb{E}[Y|I=do(i)]=\mathbb{E}[\sum_{a}P(Y|i, a)P(a)] = \mathbb{E}_A\mathbb{E} \left[ Y|i, A  \right].$ The detailed proof and derivation process can be found in Appendix C. Inspired by this theorem, we concluded that achieving an unbiased output requires inputting both the image $i$ and the attack indicator $A$ into the model, for which we use the frequency spectrum as the indicator. While using the image as input is straightforward, as most models already process images, incorporating the additional attack indicator presents a challenge. To address this, we designed an additional network, AIN, to process the attack indicator $A$ while leaving the original victim model (SFRN, e.g., ViT) unchanged. The input to the SFRN remains the image, and the encoded outputs from both networks are concatenated in the final layers for prediction. As shown in $\mathbb{E} \left[ Y|i, A \right]$, **making the final prediction requires two encoded information: the encoded information from both the attack indicator and the image**. The AIN processes $A$ and provides the encoded attack information, while the SFRN processes the image and provides the encoded image information. Both are combined before making the final prediction. Thus, the construction of the SFRN and AIN is primarily inspired by the theoretical framework of $\mathbb{E} \left[ Y|i, A \right]$.
> >
> > 2. **Empirical designs for how to better separate the learning process in AIN and SFRN**. To better encourage the AIN to capture backdoor features and suppress the SFRN to learn backdoor features, we adopt the following empirical strategies in the network design. Firstly, the AIN is intentionally designed to have a much smaller structure than the SFRN, as backdoor patterns are simpler and quicker to learn than normal patterns, as evidenced by prior work [1-4]. Thus, we use a weaker model for AIN (a 6-layer CNN) compared to the SFRN (e.g., ViT). This weaker structure inherently limits AIN's capacity to learn complex semantic features, allowing it to focus primarily on simple and easily detectable patterns. Secondly, AIN receives the frequency spectrum of the image, while SFRN processes the original image. Since the original image contains more semantic features than its frequency spectrum, this design further guides AIN to capture the trigger patterns, while SFRN learns the more complex semantic features.
> >
> > > Does the high prominence of AIN's predictions mainly dominate the reduction of learning loss on poisoned samples, thereby weakening SFRN's learning of the backdoor?
> >
> > Yes, we agree with your opinion. This is closely related to our explanations above and below.

---

> > ### Author Response · Authors · 2024-11-27
> > **Thanks for your further feedback and response to your final concern (Part 2)**
> >
> > > If this is true, for SFRN, is the confounding bias related to the backdoor (as shown in Figure 2) actually broken due to the cut-off of $A \rightarrow Y$, rather than $A \rightarrow I$
> >
> > We kindly argue that the confounding bias related to the backdoor (as shown in Figure 2) actually broken by the cutting-off of $A \rightarrow I$. Firstly, this is supported by the theoretical derivation mentioned above. Secondly, **cutting-off $A \rightarrow I$ does not mean we erase the triggers from the physical image, but that we cut off the relation between the backdoor trigger and the attack, which means this trigger is not a sign of backdoor attack but a random noise to SFRN**. From an empirical perspective, the AIN extracts the poisoned portion from the model, allowing the SFRN to primarily focus on learning semantic features rather than backdoored patterns. While the input to the SFRN may still contain trigger patterns, these patterns fail to activate the backdoor because the SFRN interprets them as noise rather than meaningful triggers of the attack (effectively cutting off $A \rightarrow I$). Moreover, the AIN captures the backdoor predictions, indicating that the path $A \rightarrow Y$ is still represented in the model through the AIN.
> >
> >
> > We hope this explanation addresses your concerns and clarifies the theoretical and empirical underpinnings of our approach.
> >
> >
> > [1] Qin Liu, Fei Wang, Chaowei Xiao, and Muhao Chen. From shortcuts to triggers: Backdoor defense with denoised poe. arXiv preprint arXiv:2305.14910, 2023
> >
> > [2] Zaixi Zhang, Qi Liu, Zhicai Wang, Zepu Lu, and Qingyong Hu. Backdoor defense via deconfounded representation learning. CVPR, 2023
> >
> > [3] Yu D, Zhang H, Chen W, et al. Indiscriminate poisoning attacks are shortcuts[J]. 2021.
> >
> > [4] Sandoval-Segura P, Singla V, Fowl L, et al. Poisons that are learned faster are more effective[C] Proceedings of the IEEE/CVF Conference on Computer Vision and Pattern Recognition. 2022: 198-205.

---

> ### Comment · Reviewer_zSrN · 2024-11-27
> **Thank you for your response**
>
> I sincerely appreciate the authors’ explanation for my final concern. While I maintain my view that the separate learning processes of AIN and SFRN depend on the empirical design of the MCCI framework, I acknowledge that the extensive experiments demonstrate the effectiveness of MCCI framework. Furthermore, the discussion surrounding this empirical design significantly enhances the clarity of the proposed method.
>
> Please ensure that all additional experiments and explanations from our discussion, which contribute to improving the method’s clarity, are incorporated into the revision.
>
> I thank the authors for their substantial efforts in the rebuttal and I have decided to raise my score.

---

> > ### Author Response · Authors · 2024-11-28
> > **Thank you for your response**
> >
> > Thank you very much for your encouragement. We fully agree with you that there are currently no provable theoretical guarantees for the empirical observations regarding SFRN. To address this, we have incorporated the previous discussion in Appendix S of the updated manuscript. We wholeheartedly agree and believe that future explorations along these lines will be both valuable and highly insightful.
> >
> > Once again, please allow me to express my deepest gratitude for your invaluable suggestions and thoughtful engagement with this work.

---

### Official Review · Reviewer_WxD1 · 2024-10-30

**Soundness:** 3
**Presentation:** 3
**Contribution:** 3
**Rating:** 6
**Confidence:** 3

**Summary:**

This paper introduces a novel approach called MCCI, designed to defend against backdoor attacks. This approach employs causal inference techniques to effectively distinguish between clean and poisoned data. The primary objective of this approach is to train a robust model from a poisoned dataset by leveraging both the images and attack indicators, thereby enhancing the model's robustness against various backdoor attack methods. Furthermore, empirical evidence demonstrates that MCCI is highly adaptable for backdoor defense in large pre-trained models while maintaining minimal training overhead. Comprehensive experiments demonstrate the effectiveness of MCCI.

**Strengths:**

1.This paper applies causal reasoning to backdoor defense with theoretical proof.
2.The proposed approach is an end-to-end method, which is suitable for generalization to large pre-trained models.
3.This paper designed a more challenging and practical threat model. The defender in the threat model expects the model to predict the original, correct labels for backdoored samples.
4.This paper conducted comprehensive ablation studies to demonstrate the effectiveness of each component in MCCI.

**Weaknesses:**

1. This paper needs to demonstrate the effectiveness of the proposed approach on different model structures, such as VGG and MobileNet.
2. This paper needs to conduct more experiments to defend against advanced backdoor attacks designed from a frequency perspective, such as LF [1].

Reference
[1]Zeng, Yi, et al. Rethinking the backdoor attacks’ triggers: A frequency perspective. ICCV’2021.

**Questions:**

Please refer to the weaknesses.

---

> ### Author Response · Authors · 2024-11-22
> **Response to Reviewer WxD1**
>
> **Q1. More experiments with VGG-16 and MobileNet**
>
> Thanks for the insightful question. According to your suggestions, we have evaluated the effectiveness of our method by using VGG16 and MobileNet as the backbone model of SFRN. We chose BadNets on the Cifar10 dataset as a simple proof-of-concept. The results are presented below:
>
>
> | Model $\downarrow$   | CA (\%) | ASR (\%)   | ARR (\%)  |
> |----------|----------|------------|------------|
> | MobileNet-v2 | 83.62 | 1.26 | 83.41 |
> | VGG-16 | 87.21  | 1.20   | 87.55  |
>
> **Q2. Low-frequency (LC) attack**
>
> We appreciate your insightful feedback and acknowledge that our defense method, while robust, is not a panacea. As you rightly pointed out, the use of the frequency spectrum as the representation of an attack may not defend against attacks specifically designed to evade detection by the Attack Indication Network (AIN) (like low-frequency attacks). In response to your suggestion, we have **included a paragraph discussing the limitations of our approach (Appendix P)** to provide a balanced view.
>
>   Moreover, we have **conducted additional adaptive attack experiments(Appendix O)** using the smooth trigger as you suggested [3]. In particular, we adopt their strategy by updating the smooth trigger with the perturbation that remains after the low-pass filter for each iteration subject to this trigger is also capable of launching a successful backdoor attack. The remaining parts of the filter perturbation can be interpreted as $t'=t*g$. Here, $t'$ is the result of the perturbation after convolving with the low-pass filter $g$ in the image domain. Hence the poisoned image can be expressed as $\hat{x}_i={x}_i+t'$. The optimization process can be written as follows:
>
> $\min_{t}   \sum_{i=1}^{|D_b|}  L(x_i+ t*g,y_{t};\theta_{p}),
> s.t. \theta_{p}=\min_{\theta} (\sum_{i=1}^{|D_{c}|} L(x_i,y_i;\theta)+ \sum_{i=1}^{|D_b|}  L( \hat{x}\_i, y_{t}; \theta) ).$
>
>   This bilevel optimization function aims to find a smooth pattern $t*g$, which falls within the range of the low-pass filter $g$, and can be adopted as a backdoor trigger to successfully compromise the model. The results in the table (see row 'Smooth Trigger') show that this smooth trigger can backdoor the model. However, the extent of the backdoor is substantially reduced after finetuning the model with a small set (25) of clean images for a few epochs (10), as shown in the last row.
>
>
>   | **Techniques** | **CA↑ (MCCI-AVG)** | **ASR↓ (MCCI-AVG)** | **ARR↑ (MCCI-AVG)** | **CA↑ (MCCI-SG)** | **ASR↓ (MCCI-SG)** | **ARR↑ (MCCI-SG)** |
>   |----------------|--------------------|---------------------|---------------------|-------------------|--------------------|--------------------|
>   | Averaging                | 84.65              | 3.32                | 64.05               | 85.98             | 3.22               | 63.20              |
>   | Gaussian            | 84.65              | 5.12                | 70.06               | 85.51             | 3.25               | 70.98              |
>   | Median             | 85.37              | 1.64                | 75.86               | 85.27             | 2.93               | 76.41              |
>   |Smooth Trigger |	84.62|	99.34|	10.00|	84.34|	99.34|	10.00|
>   |After Finetuning|82.23	|6.45|75.02	|82.52|	9.01|	75.93	|
>
>
>   Furthermore, we emphasize that the use of the frequency spectrum to indicate attacks serves as a proof of concept for our broader method of Backdoor Adjustment for Backdoor Attack. ($\mathbb{E}[Y|I=do(i)]=\mathbb{E}[\sum_{a}P(Y|i,a)P(a)] = \mathbb{E}_A\mathbb{E} \left[ Y|i, A  \right]$) This method posits that if more sophisticated input-level backdoor detection methods become available in the future, we could readily adapt our approach to incorporate these new methods in place of the frequency spectrum to represent $A$. This adaptability highlights a promising direction for future research in enhancing the robustness of models against backdoor attacks.

---

> > ### Comment · Reviewer_WxD1 · 2024-11-25
> >
> > Thanks for the reply! I recommend incorporating these discussions into the final version, as this would make the paper more comprehensive and persuasive. I will keep my score.

---

### Official Review · Reviewer_z94c · 2024-11-03

**Soundness:** 3
**Presentation:** 2
**Contribution:** 3
**Rating:** 6
**Confidence:** 4

**Summary:**

The paper proposes a defense against backdoor attacks based on data poisoning.

A standard discriminative classification architecture can be expressed as $x\mapsto c_w(f_\theta(x))$, where $f_\theta$ is a feature extractor, and $c_w$ is a classification head: $c_w(z) = \operatorname{softmax}(W z + b)$.

The defense modifies the architecture into $x\mapsto c_{w'}(f_\theta(x)\operatorname{++} a_\phi(\operatorname{DCT}(x)))$, where $\operatorname{++}$ is concatenation, $\operatorname{DCT}$ is discrete cosine transform, $a_\phi$ (AIN) is an additional lightweight feature extractor. We can express $c_{w'}(z\operatorname{++} z_\text{f}) = \operatorname{softmax}(W_1 z + W_2 z_\text{f} + b)$. Training minimizes the standard classification NLL w.r.t. to the parameters $\theta$, $\phi$, and $w'=(W_1, W_2, b)$.

In inference, the defense replaces the input of $a_\phi$ with a clean input, which causes the model to predict the clean class of the input of $f_\theta$ even if the input contains a trigger.

Some of the most important assumptions of the defense are (approximately) as follows:
1. Images with triggers have frequency spectrums distinguishable from clean images (more pronounced high-frequencies).
2. The inductive biases are such that the modified model will separate clean (semantic) features into the output of $f_\theta$ (SFRN), and trigger features into the output of $a_\phi$ (AIN).

It is intuitive (with some additional less load-bearing assumptions) that, if an intervention only replaces the input of $a_\phi$ with any clean input, $c_{w'}$ will classify the main input into its clean class.

The second assumption is confirmed empirically even though the loss function does not enforce it. That is, there exists a solution that $f_\theta$ learns both clean and trigger features, and $a_\phi$ learns nothing.

**Causal model**

The paper claims a correspondence with the causal model $\{I\to Y, A\to I, A\to Y\}$, where $I$ is the input, $A$ is the adversary, and $Y$ is the label. Then it claims that the replacement of the input of the corresponds to a causal intervention called backdoor adjustment that removes the edge $A\to I$, which breaks the confounding path $I\gets A\to Y$ and makes the prediction independent of the trigger.

**Strengths:**

- The method is simple and interesting. In inference, instead of the modified architecture, the standard classification architecture can be used. We can see this by expressing it as $x\mapsto \operatorname{softmax}(W_1 f_\theta(x) + b')$, where $b' = W_2 z_\text{f} + b$, is computed once from one or more clean inputs (using the notation from my summary).
- The experimental results are very interesting. Considering that the loss function is standard NLL, the training method is surprisingly successful in producing a model that is "blind" to trigger features and classifies inputs with triggers into their original class.
- The paper proposes evaluation of accuracy of recovery of the original label of poisoned inputs (which is analogous to robust accuracy for adversarial examples). I agree that this makes sense.
- The paper is well-structured and the quality of the writing is mostly good.

**Weaknesses:**

**Updates:**
- minor corrections,
- updated rating and soundness based on the discussion and paper updates.

The authors have addressed most of the weaknesses, including the ones that I considered most important.

---

1. Some things could be made more clear:
	1. The meaning of "(associated) attack indicator" is initially hard to understand and could be clarified earlier
	3. It is not clear whether "otherwise" in L175 is the negation of the whole antecedent or a part of it.
	4. Why does Proposition 3.4 use the expectation instead of a conditional probability distribution like in Eq. (6). Expectation seems to me to be less natural for a random variable that is a class and Eq. (6) is a more "direct", stronger statement.
	5. Definition 4.1 is not a formal definition.
	6. L258: meaning of "ideal".
	7. Fig. 4 and 5: What are DACC and ADCP?
2. The paper does not make it clear how and how well the proposed method corresponds to the causal model. I think that there are no theoretical guarantees for the assumption 2 in my summary. It seems like there is nothing preventing SFRN to learn the features that AIN learns. Perhaps a loss function that includes minimization of mutual information between the two branches of the model might make the method more robust.
3. The paper does not acknowledge the limitations of the assumptions about the backdoor attack sufficiently. The paper includes experiments with attacks that attenuate high frequencies by blurring, but the frequency spectrum remains distinguishable. I strongly suspect that attacks can be designed to have triggers that AIN does not learn. Perhaps the trigger could be more natural looking, blended and randomly positioned, or be the smooth trigger from Zeng et al. (2021), Refool by Liu et al. (2020), or something like a tennis ball [1]. Consider the a priori indistinguishability of triggers from natural features [1]. If such an attack exist and it is acknowledged clearly, I will not consider this to be a great weakness since the method still seems to be applicable to many practical triggers. I suggest devising stronger adaptive attacks.
4. The experimental evaluation should be improved:
	1. Table 1: Clean accuracies on CIFAR-10 are at most ~87%, which is about 7 percentage points lower than the original results in the ASD and DBD papers. Using stronger learning setups and clarifying the differences would make the results more valuable. Why are the standard CIFAR-10 hyperparameters not used, like in the used open-source implementations?
	2. All 4 baselines evaluate on 3 datasets, but the paper has only 2 datasets in the comparison.
	3. Related work [2] presenting a training-time defense is not cited and compared with.
5. Source code for reproducing the experiments is not provided.

To sum up, I find it most important to:
- improve clarity and relate the theory better with the machine learning algorithm,
- devise stronger adaptive attacks,
- express the limitations of the work more clearly,
- address the drawbacks of the experimental evaluation,
- provide source code for reproducing the experiments.

[1] Khaddaj et al., Rethinking Backdoor Attacks, https://arxiv.org/abs/2307.10163
[2] Liu et al., Beating Backdoor Attack at Its Own Game, https://arxiv.org/abs/2307.15539

There are some errors and potential errors (which do not affect my rating):
- L088: "(4)SOTA".
- L247-L249 "Equations equation", "Equation equation", ...
- Eq. (4): "(x))" is missing.
- "methods works".
- L460: "mages".
- The descriptions in Appendix F should are not very clear and I think that it is not correct to say that perturbing the images in LC and TUAP is not part of conducting backdoor attacks.
- L1104-1119: "(more details can be found in the Appendix)" is in the Appendix. Check the paragraphs for redundancy.

I find the paper valuable and hope that my concerns can be addressed.

I apologize for any errors on my part.

**Questions:**

(See weaknesses.)

Suggestions:
- L068: A literature reference could be added for backdoor adjustment theory.
- Fig. 3: Make poisoned examples more distinguishable and consider clarifying why Fig. 3 (c) and Fig. 3 (a) look the same.
- Consider mentioning poisoning rates in subscripts of attack names in tables.
- Include a measurements for a single clean input in figures 4 and 5 for better supporting the claim of sufficiency of a single clean frequency spectrum?
- Consider quantitatively comparing $b'$ and $b$ (see the definition under Strengths).
- Do you know what causes the drop in accuracy under the ISSBA attack?

---

> ### Author Response · Authors · 2024-11-22
> **Response to Reviewer z94c (Part 1)**
>
> **Q1. More clarifications**
>
> We sincerely appreciate your detailed review and thoughtful suggestions. Your feedback has significantly improved the clarity and quality of our paper. Following your suggestions, we **have made several changes, summarized below and highlighted in blue in the latest version**:
>
> - **Definition of Attack Indicator**. We have defined the attack indicator in the introduction to help readers better understand its role in our method.
>
> - **Clarifications in Assumptions and Definitions**. We have revised Assumption 3.3 and Definition 4.1 to make them more formal and reduce ambiguity.
>
> - **Expectation vs. Conditional Probability**. Thank you for raising this important question. We adopt the expectation rather than the conditional probability distribution because we follow the framework of causal inference to address the backdoor detection problem. Specifically: In causal inference, one of the main goals is to determine the expected outcome (treatment effect) for a given unit $X$, expressed as $\mathbb{E}[Y|x]$. To achieve this, the backdoor adjustment theory is employed to estimate outcomes free from the influence of spurious paths. This aligns naturally with the backdoor detection problem, and we leverage similar concepts and forms to frame our approach.
>
> - **Meaning of "ideal"**. Thank you for pointing out the need for additional explanation. We have expanded our discussion to clarify why the frequency spectrum is an ideal attack indicator. Specifically: An intuitive way to indicate the presence of an attack is by leveraging off-the-shelf backdoor detection algorithms [1-2]. However, these approaches depend on already **backdoored models** and require computationally intensive processes, which are impractical for representing $A$ as an input in our setting, where a clean model is trained directly from a poisoned dataset. Hence there is no backdoored model in our settings. Inspired by input-level detection methods [3], we use the frequency spectrum of each image to indicate the presence of an attack. This approach is effective due to the significant differences between backdoored and clean images in the frequency domain. Its proactive (no need an alreay backdoored model)and computationally efficient nature, along with its independence from already trained backdoored models, makes it an ideal attack indicator.
>
> - **Typo Corrections in Figures 4 and 5**. We have corrected the typographical errors in Figures 4 and 5. Specifically, DACC refers to clean accuracy, and DACP refers to the Attack Recovery Rate. We apologize for the inconsistency in naming.

---

> > ### Comment · Reviewer_z94c · 2024-11-22
> > **Clarity concerns addressed**
> >
> > Thank you!
> >
> > >**Expectation vs. Conditional Probability**.
> >
> > Apologies if I was not clear enough. My thinking was that the problem is that, because the values that $Y$  takes are from a discrete unordered set (classes), it does not make sense to multiply and add to get the expectation.
> >
> > However, now I see a way to represent classes so that $\mathbb E[Y\mid x]$ is defined: if you define $Y$ to take values in the set of one-hot indicator vectors $\bf e_i$ that represent classes $i$, then the $i$-th element of the expected vector is the probability of class $i$: $\mathbb E[Y\mid x][i]=P(Y=\bf e_i\mid x)$.
> >
> > >**Meaning of "ideal".**
> >
> > I understand now that "ideal" is used somewhat in a colloquial way, not as something that is optimal in some sense. I do not think the usage of "ideal" in the paper is very useful other than for distinguishing the proposed method from others because the criteria that the paper chooses are not general principles, but only what the proposed method is good at. However, I do not consider this point to be very important.
> >
> > I consider the points under Q1 to be addressed sufficiently well.

---

> > > ### Author Response · Authors · 2024-11-22
> > > **Thanks for your further feedback**
> > >
> > > Thank you for your additional feedback. We are glad that most of your concerns of Q1 have been addressed. We also appreciate your clarifications regarding expectation and conditional probability. We found this point very insightful and have included it in our appendix highlighted in blue. Furthermore, we understand your concern about the use of the term 'ideal,' which might cause confusion. Consequently, we have replaced it with 'a highly appropriate choice' in the latest version of our manuscript.

---

> ### Author Response · Authors · 2024-11-22
> **Response to Reviewer z94c (Part 2)**
>
> **Q2. Clarifications on the correlation of our method and the causal model**
>
> We sincerely appreciate your question regarding the correlation between our method and the causal model. We would like to provide further clarification on these associations.
>
> To propose a defense method capable of training a clean model on a poisoned dataset, it was first necessary to understand the intrinsic differences between training a poisoned model and a clean model under such conditions. Specifically, we utilized causal inference and constructed a causal model, as you summarized in the review. Our analysis revealed that the underlying issue leading to the poisoned model is the presence of a spurious path in the causal model. Building on this understanding, we sought an approach to eliminate this spurious path. Inspired by the backdoor adjustment technique in causal inference—originally designed to address such spurious paths—we framed our problem and developed our method based on this theoretical foundation.
>
> In our work, there is a straightforward way to ensure Assumption 2 in your summary ("The inductive biases are such that the modified model will separate clean (semantic) features into the output of SFRN and trigger features into the output of AIN"), as illustrated in Remark 4.5 (Design Guidelines for AIN). Specifically, the AIN is intentionally designed to have a much smaller structure than the SFRN, as backdoor patterns are simpler and quicker to learn than normal patterns, as evidenced by prior work [6-9]. Thus, we use a weaker model for AIN (a 6-layer CNN) compared to the SFRN (e.g., ViT). This weaker structure inherently limits AIN's capacity to learn complex semantic features, allowing it to focus primarily on simple and easily detectable patterns. Additionally, the inputs to AIN and SFRN differ: AIN receives the frequency spectrum of the image, while SFRN processes the original image. Since the original image contains more semantic features than its frequency spectrum, this design further guides AIN to capture the trigger patterns, while SFRN learns the more complex semantic features. We also find your suggestion of adding a mutual information loss between the embeddings from AIN and SFRN intriguing, as it could promote greater independence between the two. However, implementing this approach may require introducing an additional model [10] to approximate the mutual information during training, adding complexity to the method. Given that our approach is designed with large pre-trained models in mind, incorporating additional structures and losses could make the method cumbersome. Since the current approach already achieves SOTA performance, we opted to maintain its simplicity.

---

> > ### Comment · Reviewer_z94c · 2024-11-22
> > **Empirical validation of correspondence between the causal model and the trained model is good, but lack of theoretical guarantee is not acknowledged**
> >
> > Thank you for the clarifications!
> >
> > > [...] we framed our problem and developed our method based on this theoretical foundation
> >
> > I agree that this makes sense.
> >
> > > In our work, there is a straightforward way to ensure Assumption 2 in your summary
> >
> > This makes intuitive sense to me. However, I wouldn't be able to guess that the method would work so well with the NLL loss. What I find particularly interesting is the empirical observation that SFRN doesn't learn trigger features despite being able to learn them easily when there is no AIN. I would not be confident about this before seeing the experiments. (Let me know if you disagree.)
> >
> > My concern is that there is no theory that implies this from the properties of the learning algorithm – the causal model only assumes my Assumption 2. Hence, I don't agree that Assumption 2 is ensured.  However, I find the empirical observation very interesting and think that acknowledging the lack of theoretical guarantees would improve the paper.
> >
> > > Since the current approach already achieves SOTA performance, we opted to maintain its simplicity.
> >
> > I agree that this is reasonable.

---

> > > ### Author Response · Authors · 2024-11-23
> > > **Thanks for your further feedback**
> > >
> > > Thank you for your further response. Although the method already achieves good separation and performance, as demonstrated by the experiments, we acknowledge that providing additional theoretical guarantees along with corresponding loss functions would be a valuable improvement. Therefore, we have added this point to the limitations and future work section in the latest version of our paper.

---

> ### Author Response · Authors · 2024-11-22
> **Response to Reviewer z94c (Part 3)**
>
> **Q3. Potential limitations of MCCI**
>
> We appreciate your insightful feedback and acknowledge that our defense method, while robust, is not a panacea. As you rightly pointed out, the use of the frequency spectrum as the representation of an attack may not defend against attacks specifically designed to evade detection by the Attack Indication Network (AIN) (like low-frequency attack). In response to your suggestion, we have **included a paragraph discussing the limitations of our approach (Appendix P)** to provide a balanced view.
>
>   Moreover, we have **conduced additional adaptive attack experiments(Appendix O)** using the smooth trigger as you suggested [3]. In particular, we adopt their strategy by updating the smooth trigger with the perturbation that remains after the low-pass filter for each iteration subject to this trigger is also capable of launching successful backdoor attack. The remaining parts of the filter perturbation can be interpreted as $t'=t*g$. Here, $t'$ is the result of the perturbation after convolving with the low-pass filter $g$ in the image domain. Hence the poisoned image can be expressed as $\hat{x}_i={x}_i+t'$. The optimization process can be written as follows:
>
> $\min_{t}   \sum_{i=1}^{|D_b|}  L(x_i+ t*g,y_{t};\theta_{p}),
> s.t. \theta_{p}=\min_{\theta} (\sum_{i=1}^{|D_{c}|} L(x_i,y_i;\theta)+ \sum_{i=1}^{|D_b|}  L( \hat{x}\_i, y_{t}; \theta) ).$
>
> This bilevel optimization function aims to find a smooth pattern $t*g$, which falls within the range of the low-pass filter $g$, and can be adopted as a backdoor trigger to successfully compromise the model. The results in the table (see row 'Smooth Trigger') show that this smooth trigger can backdoor the model. However, the extent of the backdoor is substantially reduced after finetuning the model with a small set (25) of clean images for a few epochs (10), as shown in the last row.
>
>
>   | **Techniques** | **CA↑ (MCCI-AVG)** | **ASR↓ (MCCI-AVG)** | **ARR↑ (MCCI-AVG)** | **CA↑ (MCCI-SG)** | **ASR↓ (MCCI-SG)** | **ARR↑ (MCCI-SG)** |
>   |----------------|--------------------|---------------------|---------------------|-------------------|--------------------|--------------------|
>   | Averaging                | 84.65              | 3.32                | 64.05               | 85.98             | 3.22               | 63.20              |
>   | Gaussian            | 84.65              | 5.12                | 70.06               | 85.51             | 3.25               | 70.98              |
>   | Median             | 85.37              | 1.64                | 75.86               | 85.27             | 2.93               | 76.41              |
>   |Smooth Trigger |	84.62|	99.34|	10.00|	84.34|	99.34|	10.00|
>   |After Finetuning|82.23	|6.45|75.02	|82.52|	9.01|	75.93	|
>
>
>   Furthermore, we emphasize that the use of the frequency spectrum to indicate attacks serves as a proof of concept for our broader method of Backdoor Adjustment for Backdoor Attack. ($\mathbb{E}[Y|I=do(i)]=\mathbb{E}[\sum_{a}P(Y|i,a)P(a)] = \mathbb{E}_A\mathbb{E} \left[ Y|i, A  \right]$) This method posits that if more sophisticated input-level backdoor detection methods become available in the future, we could readily adapt our approach to incorporate these new methods in place of the frequency spectrum to represent $A$. This adaptability highlights a promising direction for future research in enhancing the robustness of models against backdoor attacks.
>
> **Q4. More Clarification on experimental setups**
>
> **Q4.1 Clarifications on the lower clean accuracy**
>
> The details of our training recipe are provided in Appendix I. It is noted that the main reason for the performance gap is that we didn't apply any data augmentations in the experiments, such as random rotation, or shifting. This choice follows the rationale in the ABL. Specifically, data augmentations potentially hinder the backdoor effect [14].

---

> > ### Comment · Reviewer_z94c · 2024-11-23
> > **Good new adaptive attack experiments, weak baseline setup (from ABL) for comparison of methods**
> >
> > Thank you! I appreciate your effort.
> >
> > > **Q3. Potential limitations of MCCI**
> >
> > The experiments are interesting. Please make sure to update the experiments section in the final paper.
> >
> > How does the fine tuning work? How are the 25 clean images selected (what is their class)?
> >
> > > **Q4.1 Clarifications on the lower clean accuracy**
> >
> > [DBD (Appendix A)](https://arxiv.org/abs/2202.03423) and [ASD (Appendix B.2)](https://arxiv.org/abs/2303.12993) use the following setup for CIFAR-10:
> > > we use the SGD optimizer with momentum $0.9$, weight decay of $5 \times 10^{−4}$ , and an initial learning rate of $0.1$. The batch size is set to $128$ and we train the ResNet-$18$ model $200$ epochs. The learning rate is decreased by a factor of $10$ at epoch $100$ and $150$, respectively.
> >
> > The data augmentation is the standard padding, cropping and horizontal flipping from [A].
> >
> > It seems that the most important difference is that your paper and [ABL (section 4)](https://arxiv.org/abs/2110.11571) use a non-decaying learning rate and no data augmentation (for some attacks). (Please check whether you have horizontal flipping, which seems to be present in the source code.)
> >
> > I consider the DBD and ASD setup to be superior because it makes it harder to improve accuracy by changing hyperparameters unrelated to the defense method (comparisons are controlled better). Therefore, I think that the lack of experiments in this setup is an important weakness, but I would like to know what you and other reviewers think.
> >
> > ---
> >
> > [A] He et al. (2016). Identity Mappings in Deep Residual Networks. https://www.arxiv.org/abs/1603.05027

---

> > > ### Author Response · Authors · 2024-11-23
> > > **Thanks for your further feedback**
> > >
> > > Thanks so much for your responses and encouragement on our new adaptive results.
> > >
> > > We have already added those experiments in our latest paper in Appendix O.
> > >
> > > Regarding Q3, our intuition for adopting fine-tuning strategy is driven by the previous well-established works [1, 2]. Specifically, fine-tuning the DNN on purely clean samples potentially leads the model to forget about backdoor information. Moreover, the 25 samples are randomly chosen from the clean training set.
> > >
> > > Regarding Q4.1,
> > >
> > > Firstly, we'd like to add that we did use a 'decaying learning rate' in the experiments. The learning rate is scheduled to decrease by a factor of 10 at epochs 30, 60, and 90. Sorry about the inconsistencies in our paper writing, we have updated the paper accordingly.
> > >
> > > Secondly, we would like to clarify that **this experimental setup (i.e., no data augmentations) is applied uniformly across all baseline methods, not just MCCI**. Our rationale is that the effectiveness of attack methods might be potentially influenced if data augmentations are introduced in the training stage [3]. Therefore, evaluating these defense methods in the scenario of no data augmentations makes the attack more challenging for the defense.
> > >
> > > [1] Liu K, Dolan-Gavitt B, Garg S. Fine-pruning: Defending against backdooring attacks on deep neural networks. InInternational symposium on research in attacks, intrusions, and defenses 2018 Sep 7 (pp. 273-294). Cham: Springer International Publishing.
> > >
> > > [2] Li Y, Lyu X, Koren N, Lyu L, Li B, Ma X. Neural attention distillation: Erasing backdoor triggers from deep neural networks. arXiv preprint arXiv:2101.05930. 2021 Jan 15.
> > >
> > > [3] Liu, Y., Ma, X., Bailey, J., & Lu, F. (2020). Reflection backdoor: A natural backdoor attack on deep neural networks. In Computer Vision–ECCV 2020: 16th European Conference, Glasgow, UK, August 23–28, 2020, Proceedings, Part X 16 (pp. 182-199). Springer International Publishing.

---

> ### Author Response · Authors · 2024-11-22
> **Response to Reviewer z94c (Part 4)**
>
> **Q4.2 Evaluating MCCI on more datasets**
>
> Thanks for the question. Firstly, we'd like to argue that the choice of datasets follows the previous well-established works [11-12] in machine learning safety and trustworthy AI. But to further demonstrate the effectiveness of our method, we compare our method with the baselines on the popular GTSRB dataset. The results are shown below:
>
> | GTSRB  | ABL (\%)    | CBD (\%)  | DBD (\%)        | ASD  (\%)  | Ours-avg (\%) |
> |--------|--------|-------|-------------|--------|------------|
> | BadNets| 97.22, 0.58, 97.13  | 92.19, 0.03, 95.40 | 88.21, 0.00, 	89.23| 97.04, 0.05, 94.35 | 97.57, 0.00, 95.23    |
> | Blend  | 81.13, 29.96, 37.6 | 91.25, 0.3, 89.61| 88.23, 1.00, 0.00  | 97.25, 4.65, 95.57  | 97.62, 0.00, 94.36|
> | WaNet  | 96.36, 79.17, 19.54 | 92.13, 29.95, 37.69| 90.06, 0.00,	89.56| 97.27, 4.02, 97.37  | 96.27, 0.15, 88.98 |
> | ISSBA  | 89.78, 11.64, 69.61 | 81.13, 8.22, 69.61| 82.23, 	100, 0.00 | 97.23, 3.51, 6.41| 97.34, 0.00, 96.46|
>
> It is noted that each cell contains three values, which correspond to clean accuracy (CA), attack success rate (ASR), and attack recovery rate (ARR). It is shown that our method consistently shows good performance in recovering original labels for backdoor samples, while also maintaining a good clean accuracy and a low attack success rate.
>
> **Q4.3 Comparison with [13]**
>
> Thank you for bringing this intreseting related work to our attention. We have cited and compared it in the related work section. Moreover, we have added additional experiments of it on CIFAR10, the results can be shown in below. However, we respectfully argue that it may not fulfill our threat model. This method proposes adding a backdoor $t'$, whose target label is its original label, to the current backdoor $t$. In the inference stage, adding this $t'$ to every sample supposedly suppresses backdoored predictions. To achieve this, the method first requires an **additional backdoor detection tool** to isolate backdoored samples and an **advanced predictor** to identify the original labels of these samples. However, in our setting, defenders do not have access to these additional tools. We also contend that assuming the availability of such tools is impractical for defending against backdoors in large pre-trained models due to the vast amount of training data and the lack of comparably effective prediction models.
>
> | Method   | NAB CA (\%) /ASR (\%)  /ARR (\%)  | Ours-AVG CA (\%) /ASR (\%)  /ARR (\%)
> |----------|------------------------------|--------------------------|
> | BadNets  | 87.20 , 1.42 , 87.20     |86.73 , 0.50 , 85.64|
> | Blend    | 87.33 , 11.10, 84.84  |85.83 , 1.94 , 80.05 |
> | WaNet    | 87.49 , 1.43 ,  86.39|81.04 , 15.54 , 75.98 |
>
> It is noticed that NAB achieves a higher clean accuracy compared to our method. The advantages might be attributed to the additional backdoor detection tool and the advanced predictor introduced by NAB.
>
> **Q5. Source code issues**
>
> We have now published our code. You can find it here https://anonymous.4open.science/status/BKD_BKD_ICLR-503E.
>
>
> Thank you for your detailed review and for pointing out the typos. We have corrected all the typos you identified in the latest version.

---

> > ### Comment · Reviewer_z94c · 2024-11-24
> > **Thank you for the additional experiments and discussion**
> >
> > Thank you for the additional experiments and discussion! This addresses my points Q4.2-Q5. I hope that the final version of the source code will enable easy reproduction of the experiments.
> >
> > ---
> >
> > **Increased rating and remaining weaknesses**
> >
> > I am increasing my score from "reject" to "weak reject".
> >
> > I would be confidently in favor of acceptance if the following weaknesses were addressed (irrespective of the experimental results):
> > 1. The paper does not include experiments with data augmentations that are necessary for higher clean accuracy. Such experiments would make the paper more useful for comparison with some other methods (ASD, DBD, D-ST, future work) and would answer some questions about how data augmentations affect the effectiveness of the method (see the discussion with reviewer zSrN).
> > 2. The text currently comes across as a bit overly defensive, including in the limitations section. It gives the impression that the effectiveness of MCCI is grounded more firmly in theory than it is. Additionally, the empirical contribution demonstrating that SFRN does not learn backdoor features is understated and could be highlighted more effectively.
> >
> > I find the paper valuable and I am sorry if the first point is too hard to address.
> >
> > ---
> >
> > Additional minor suggestions:
> > - Some paragraphs could be split into smaller ones.
> > - Correct Eq (11): `s.t.` -> `\text{ s.t. }`, `\min \theta` -> `\argmin \theta`.
> > - Correct \citet and \citep, missing space after ".", and hyperlinks in the blue text, and re-check the whole paper.

---

> > > ### Author Response · Authors · 2024-11-27
> > > **Thanks for the discussion and looking forward to your future responses**
> > >
> > > Dear Reviewer z94c,
> > >
> > > Thank you for your meticulous review and encouragement of our contribution to the scientific research community! It is precisely because of your rigorous and conscientious reviewing spirit that our research community can continue to flourish. As we approach the deadline for the final PDF modification of our article, we hope to take this last opportunity to understand if you have any remaining confusion about our paper. We will strive to address your questions.
> > >
> > > Thank you!
> > >
> > > Best Regards,
> > >
> > > Authors of Submission 4200

---

> > > > ### Comment · Reviewer_z94c · 2024-11-27
> > > >
> > > > Dear authors,
> > > >
> > > > thank you for your effort in improving the paper and the discussion. I am very much in agreement with Reviewer z94c's conclusion. I hope that you will be able to update the paper on time.
> > > >
> > > > As my concerns have been mostly addressed by the experiments and the discussion, I will raise my score.
> > > >
> > > > Best Regards,
> > > > Reviewer z94c

---

> > > > > ### Author Response · Authors · 2024-11-28
> > > > > **Thank you for your response!**
> > > > >
> > > > > Thank you very much for your encouragement. We sincerely apologize for the misunderstanding in our previous response. We fully agree with you and Reviewer zSrN that there are currently no provable theoretical guarantees for the empirical observations regarding SFRN. We believe this phenomenon is likely driven by a combination of theoretical insights and intentional empirical design. To address this, we have provided additional clarification and discussed this limitation in Appendix S of the updated manuscript.
> > > > >
> > > > > As you thoughtfully suggested, "Noticing this seems like an interesting starting point for further investigation." We wholeheartedly agree and believe that future explorations along these lines will be both valuable and highly insightful.
> > > > >
> > > > > Once again, please allow me to express my deepest gratitude for your invaluable suggestions and thoughtful engagement with this work.

---

> > > > > > ### Comment · Reviewer_z94c · 2024-11-28
> > > > > >
> > > > > > I am happy to have collaborated with you.
> > > > > >
> > > > > > Also, for the camera-ready version, I encourage you to update the "Resistance to Potential Adaptive Attack" paragraph with information from Appendix P, and fix the last sentence in the conclusion.

---

> > > > > > > ### Author Response · Authors · 2024-11-28
> > > > > > > **Thank you for your response!**
> > > > > > >
> > > > > > > Thanks again for your suggestion! We will definitely add these two modifications!

---

> ### Author Response · Authors · 2024-11-22
> **Response to Reviewer z94c (Part 5)**
>
> [1] Qi X, Xie T, Wang J T, et al. Towards a proactive {ML} approach for detecting backdoor poison samples[C]//32nd USENIX Security Symposium (USENIX Security 23). 2023: 1685-1702.
>
> [2] Minzhou Pan, Yi Zeng, Lingjuan Lyu, Xue Lin, and Ruoxi Jia. {ASSET}: Robust backdoor data detection across a multiplicity of deep learning paradigms. In 32nd USENIX Security Symposium (USENIX Security 23), pp. 2725–2742, 2023
>
> [3] Yi Zeng, Won Park, Z Morley Mao, and Ruoxi Jia. Rethinking the backdoor attacks’ triggers: A frequency perspective. In Proceedings of the IEEE/CVF international conference on computer vision, pp. 16473–16481, 2021.
>
> [4] Shalit U, Johansson F D, Sontag D. Estimating individual treatment effect: generalization bounds and algorithms[C]//International conference on machine learning. PMLR, 2017: 3076-3085.
>
> [5] Yao L, Chu Z, Li S, et al. A survey on causal inference[J]. ACM Transactions on Knowledge Discovery from Data (TKDD), 2021, 15(5): 1-46.
>
> [6] Qin Liu, Fei Wang, Chaowei Xiao, and Muhao Chen. From shortcuts to triggers: Backdoor defense with denoised poe. arXiv preprint arXiv:2305.14910, 2023
>
> [7] Zaixi Zhang, Qi Liu, Zhicai Wang, Zepu Lu, and Qingyong Hu. Backdoor defense via deconfoundedrepresentation learning. CVPR, 2023
>
> [8] Yu D, Zhang H, Chen W, et al. Indiscriminate poisoning attacks are shortcuts[J]. 2021.
>
> [9] Sandoval-Segura P, Singla V, Fowl L, et al. Poisons that are learned faster are more effective[C]//Proceedings of the IEEE/CVF Conference on Computer Vision and Pattern Recognition. 2022: 198-205.
>
> [10] Hjelm R D, Fedorov A, Lavoie-Marchildon S, et al. Learning deep representations by mutual information estimation and maximization[J]. arXiv preprint arXiv:1808.06670, 2018.
>
> [11] Guo J, Li Y, Chen X, Guo H, Sun L, Liu C. Scale-up: An efficient black-box input-level backdoor detection via analyzing scaled prediction consistency. arXiv preprint arXiv:2302.03251. 2023 Feb 7.
>
> [12] ZeroMark: Towards Dataset Ownership Verification without the Verification Watermark Disclosure
>
> [13] Liu et al., Beating Backdoor Attack at Its Own Game, https://arxiv.org/abs/2307.15539
>
> [14] Yunfei Liu, Xingjun Ma, James Bailey, and Feng Lu. Reflection backdoor: A natural backdoor attack on deep neural networks. In ECCV, 2020.

---

> ### Author Response · Authors · 2024-11-25
> **Thanks for your further feedback**
>
> **Q1. Experiments with data augmentations**
>
> Thanks for the great suggestions! To align our experimental setup with the existing advanced baseline methods (e.g, DBD, ASD, D-ST), we add two additional data augmentations (random cropping and random horizontal flipping) in the training process for **all the baselines and our methods**. The choice of these two data augmentation operations follows that in ASD (please check https://github.com/KuofengGao/ASD/blob/main/config/baseline_asd.yaml for more details). The other experimental settings (e.g., learning rate, optimizer, # epoch, etc.) are unchanged. We compare the effectivenss of our method with the baselines on the CIFAR-10 dataset. The following table presents the resutls:
>
>
> |  | ABL(\%)  | CBD(\%)   | D-ST (\%) | DBD (\%)  | ASD  (\%) | Ours-avg (\%) |
> |--------|-------------|--------|------------|------------|-------------|-------------|
> | BadNets| 89.05, 1.55, 89.63| 89.39, 1.08, 89.25 | 83.16, 14.25, 82.25 | 92.21, 1.23, 91.78| 92.69, 0.88, 91.69  | 92.33, 1.34, 92.55   |
> | Blend  | 87.48, 7.15, 68.83 | 89.95, 5.62, 90.07| 84.25, 80.05, 17.20  | 92.18, 7.49, 91.42  | 92.77, 1.23, 91.05  | 91.83, 1.82, 91.68|
> | WaNet  | 89.57, 1.41, 87.36| 80.21, 29.67, 74.79| 79.02, 13.20, 69.20  | 90.25, 0.25, 82.92|91.53, 2.73, 89.04  | 91.62, 5.24, 92.56 |
> | ISSBA  | 85.88, 5.12, 82.70 | 76.83, 91.02, 5.76 | 69.25, 68.26, 21.25 | 82.37, 0.48, 79.25  | 91.02, 3.76, 24.23  | 91.22, 1.06, 91.05 |
>
> Note that each cell contains three values: clean accuracy (CA), attack success rate (ASR), and attack recovery rate (ARR). It is shown that **data augmentations can boost clean accuracy of our method, while also maintaining its ability to achieve low ASR and consistently high ARR**. Therefore, **MCCI demonstrates robustness to data augmentations during the training stage**, as evidenced by its strong performance in both data-augmented and non-data-augmented scenarios.
>
>
> **Q2. Overly defensive tone**
>
> We appreciate you pointing out the overly defensive tone, as we also believe it potentially leads to misunderstanding about the contribution of our work. In this paper, we may have devoted too much space to describing the theoretical framework while dedicating relatively less attention to highlighting the empirical contributions. Specifically, our method not only prevents target label predictions but also effectively and robustly recovers the original, correct labels for backdoored images while maintaining high clean accuracy—a unique feature absent in previous models, as supported by extensive experiments. To address this, we have revised the abstract, introduction, and limitations sections to emphasize this aspect (highlighted in blue). We have also adjusted the order of the contributions to bring this point to the forefront. Additionally, we have revised the methods section to underscore the empirical contributions more clearly.
> Thank you for your detailed review. We also noticed that the newly added paragraph in Appendix P was too long, so we have separated it for clarity. Furthermore, we corrected the citation and link typos and made necessary updates to the terms in the equations.

---

> > ### Comment · Reviewer_z94c · 2024-11-27
> > **Thank you for the feedback!**
> >
> > Thank you! I am happy that you succeeded in performing the data augmentations experiments. Incorporating this into the paper addresses this concern.
> >
> > I also appreciate your effort towards making the paper more clear and less misleading. However, I think that you misunderstood the sentence:
> > > Additionally, the empirical contribution demonstrating that SFRN does not learn backdoor features is understated and could be highlighted more effectively.
> >
> > Reviewer zSrN refers to this as well under "Explanation of empirical observation of SFRN's learning".
> >
> > See also my paragraph that ends with:
> > > I would not be confident about this before seeing the experiments.
> >
> > While we can hypothesize that AIN is able to learn the backdoor features very fast, and SFRN is therefore less encouraged to learn backdoor features, I do not think that we have an as accurate understanding of the optimization dynamics to be able to predict this before seeing the empirical evidence. Noticing this seems like an interesting starting point for further investigation.

---

### Author Response · Authors · 2024-11-28
**To All Reviewers**

Dear Reviewers,

We sincerely thank all the reviewers for your detailed comments and constructive suggestions. We are glad that your concerns were progressively addressed throughout the discussion period. We have prepared a final version of the manuscript, incorporating all the new experimental results and discussions that took place here.

- Experiments with the additional GTSRB dataset (Appendix R.1)
- Experiments with additional data augmentations (Appendix R.2)
- Experiments including more defense baselines D-ST and NAB (Appendix R.3)
- Experiments with additional model structures (Appendix R.4)
- Detailed experimental setups for attack and defense baselines (Appendices G–J)
- Clearer statements about limitations (Appendix Q)
- Expanded and more comprehensive discussions of related literature (Related Works)
- Analysis of the empirical observations on SFRN and AIN (Appendix S)
- Other updates and modifications made during the discussions.

Finally, we would like to express our heartfelt gratitude to all the reviewers once again. Your efforts in reviewing the paper are greatly appreciated! It is thanks to your active discussions and valuable suggestions that our manuscript has been gradually improved.

Best regards,

Authors of Paper 4200

---

### Meta-Review · Area_Chair_8DzM · 2024-12-19

**Metareview:**

The method is simple and interesting, using a standard classification architecture for inference while computing certain values from clean inputs. The experimental results are impressive, with the training method successfully producing a model that ignores trigger features and classifies poisoned inputs correctly. The paper's evaluation of recovery accuracy for poisoned inputs, akin to robust accuracy for adversarial examples, makes sense. Overall, the paper is well-structured, and the writing quality is mostly good. The proposed threat model is more practical, requiring end-to-end training and original labels even for poisoned images. Their novel use of causal inference treats backdoor attacks as confounders, with defense framed as do-calculus, where the model is trained with both the image and attack indicator. Asking the model for original labels is seen as fixing the confounder to 0, enabling clean predictions when the attack indicator is 0.

**Additional Comments On Reviewer Discussion:**

It received ratings of 8, 6, 6, 6. After the discussion period most concerns of the reviewers were addressed and some of them increased their score. One of the reviewers did not change the rating in the end due to the following concern: The use of causal graphs and frequency transformations in prior defenses, along with MCCI's limited robustness against advanced attacks like LF, highlights its shortcomings compared to state-of-the-art methods that maintain clean accuracy while providing robust defense.

Since no one suggests rejecting it, it will be accepted.

---

### Decision · Program_Chairs · 2025-01-22

Accept (Poster)